# SemanticAdv: Generating Adversarial Examples via Attribute-conditional Image Editing

## Abstract

Deep neural networks (DNNs) have achieved great success in various applications due to their strong expressive power. However, recent studies have shown that DNNs are vulnerable to adversarial examples which are manipulated instances targeting to mislead DNNs to make incorrect predictions. Currently, most such adversarial examples try to guarantee "subtle perturbation" by limiting the $L_p$ norm of the perturbation. In this paper, we aim to explore the impact of semantic manipulation on DNNs predictions by manipulating the semantic attributes of images and generate "unrestricted adversarial examples". In particular, we propose an algorithm *SemanticAdv* which leverages disentangled semantic factors to generate adversarial perturbation by altering controlled semantic attributes to fool the learner towards various "adversarial" targets. We conduct extensive experiments to show that the semantic based adversarial examples can not only fool different learning tasks such as face verification and landmark detection, but also achieve high targeted attack success rate against *real-world black-box* services such as Azure face verification service based on transferability. To further demonstrate the applicability of *SemanticAdv* beyond face recognition domain, we also generate semantic perturbations on street-view images. Such adversarial examples with controlled semantic manipulation can shed light on further understanding about vulnerabilities of DNNs as well as potential defensive approaches.

## 1 Introduction

Deep neural networks (DNNs) have demonstrated great successes in advancing the state-of-the-art performance of discriminative tasks (Krizhevsky et al., 2012; Goodfellow et al., 2016; He et al., 2016; Collobert & Weston, 2008; Deng et al., 2013; Silver et al., 2016). However, recent research found that DNNs are vulnerable to adversarial examples which are carefully crafted instances aiming to induce arbitrary prediction errors for learning systems. Such adversarial examples containing small magnitude of perturbation have shed light on understanding and discovering potential vulnerabilities of DNNs (Szegedy et al., 2013; Goodfellow et al., 2014b; Moosavi-Dezfooli et al., 2016; Papernot et al., 2016; Carlini & Wagner, 2017; Xiao et al., 2018b;c;a; 2019). Most existing work focused on constructing adversarial examples by adding $\mathcal{L}_p$ bounded pixel-wise perturbations (Goodfellow et al., 2014b) or spatially transforming the image (Xiao et al., 2018c; Engstrom et al., 2017) (e.g., in-plane rotation or out-of-plane rotation). Generating unrestricted perturbations with semantically meaningful patterns is an important yet under-explored field.

At the same time, deep generative models have demonstrated impressive performance in learning disentangled semantic factors through data generation in an unsupervised (Radford et al., 2015; Karras et al., 2018; Brock et al., 2019) or weakly-supervised manner based on semantic attributes (Yan et al., 2016; Choi et al., 2018). Empirical findings in (Yan et al., 2016; Zhu et al., 2016a; Radford et al., 2015) demonstrated that a simple linear interpolation on the learned image manifold can produce smooth visual transitions between a pair of input images.

In this paper, we introduce a novel attack *SemanticAdv* which generates unrestricted perturbations with semantically meaningful patterns. Motivated by the findings mentioned above, we leverage an attribute-conditional image editing model (Choi et al., 2018) to synthesize adversarial examples by interpolating between source and target images in the feature-map space. Here, we focus on changing a single attribute dimension to achieve adversarial goals while keeping the generated adversarial image reasonably-looking (e.g., see Figure 1). To validate the effectiveness of the proposed attack method, we consider two tasks, namely, face verification and landmark detection, as face recognition field has been extensively explored and the commercially used face models are relatively robust

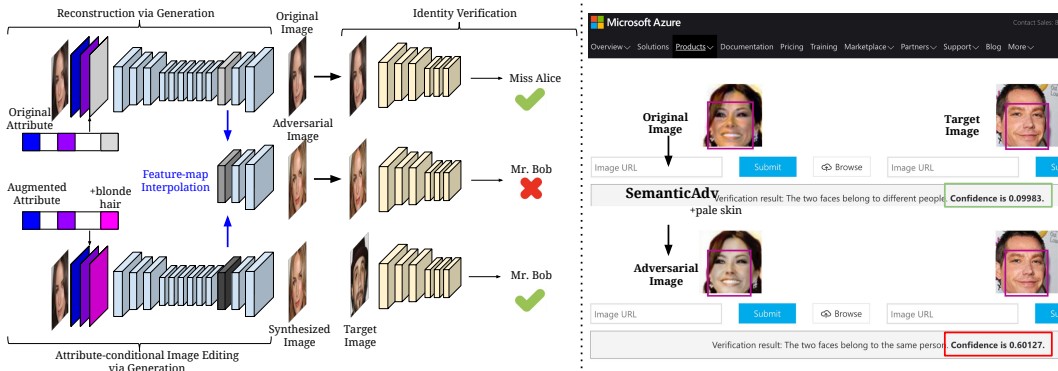

Figure 1: Left: Overview of the proposed *SemanticAdv*. Right: Illustration of our *SemanticAdv* in the real world face verification platform. Note that the confidence denotes the likelihood that two faces belong to the same person.

since they require a low false positive rate. We conduct both qualitative and quantitative evaluations on CelebA dataset (Liu et al., 2015). To demonstrate the applicability of *SemanticAdv* beyond face domain, we further extend *SemanticAdv* to generate adversarial street-view images. We treat semantic layouts as input attributes and use the image editing model (Hong et al., 2018) pre-trained on Cityscape dataset (Cordts et al., 2016). Please find more visualization results on the anonymous website: `https://sites.google.com/view/generate-semantic-adv-example`.

The **contributions** of the proposed *SemanticAdv* are three-folds. First, we propose a novel semantic-based attack method to generate unrestricted adversarial examples by **feature-space** interpolation. Second, the proposed method is able to generate **semantically-controllable** perturbations due to the attribute-conditioned modeling. This allows us to analyze the robustness of a recognition system against different types of semantic attacks. Third, as a side benefit, the proposed attack exhibits high transferability and leads to a 65% query-free black-box attack success rate on a real-world face verification platform, which outperforms the pixel-wise perturbations in attacking existing defense methods.

## 2 Related Work

**Semantic image editing.**     Semantic image synthesis and manipulation is a popular research topic in machine learning, graphics and vision. Thanks to recent advances in deep generative models (Kingma & Welling, 2014; Goodfellow et al., 2014a; Oord et al., 2016) and the empirical analysis of deep classification networks (Krizhevsky et al., 2012; Simonyan & Zisserman, 2014; Szegedy et al., 2015), past few years have witnessed tremendous breakthroughs towards high-fidelity pure image generation (Radford et al., 2015; Karras et al., 2018; Brock et al., 2019), attribute-to-image generation (Yan et al., 2016; Choi et al., 2018), text-to-image generation (Mansimov et al., 2015; Reed et al., 2016; Van den Oord et al., 2016; Odena et al., 2017; Zhang et al., 2017; Johnson et al., 2018), and image-to-image translation (Isola et al., 2017; Zhu et al., 2017; Liu et al., 2017; Wang et al., 2018b; Hong et al., 2018).

**Adversarial examples.**     Generating $L_p$ bounded adversarial perturbation has been extensively studied recently (Szegedy et al., 2013; Goodfellow et al., 2014b; Moosavi-Dezfooli et al., 2016; Papernot et al., 2016; Carlini & Wagner, 2017; Xiao et al., 2018b). To further explore diverse adversarial attacks and potentially help inspire defense mechanisms, it is important to generate the so-called "unrestricted" adversarial examples which contain unrestricted magnitude of perturbation while still preserve perceptual realism Brown et al. (2018). Recently, Xiao et al. (2018c); Engstrom et al. (2017) propose to spatially transform the image patches instead of adding pixel-wise perturbation, while such spatial transformation does not consider semantic information. Our proposed *semanticAdv* focuses on generating unrestricted perturbation with semantically meaningful patterns guided by visual attributes.

Relevant to our work, Song et al. (2018) proposed to synthesize adversarial examples with an unconditional generative model. Bhattad et al. (2019) studied semantic transformation in only the color or texture space. Compared to these works, *semanticAdv* is able to generate adversarial examples in a controllable fashion using specific visual attributes by performing manipulation in the

feature space. We further analyze the robustness of the recognition system by generating adversarial examples guided by different visual attributes. Concurrent to our work, Joshi et al. (2019) proposed to generate semantic-based attacks against a restricted binary classifier, while our attack is able to mislead the model towards arbitrary adversarial targets. They conduct the manipulation within the attribution space which is less flexible and effective than our proposed feature-space interpolation.

# 3 SEMANTIC ADVERSARIAL EXAMPLES

## 3.1 PROBLEM DEFINITION

Let $\mathcal{M}$ be a machine learning model trained on a dataset $\mathcal{D} = \{(\mathbf{x}, \mathbf{y})\}$ consisting of image-label pairs, where $\mathbf{x} \in \mathbb{R}^{H \times W \times D_I}$ and $\mathbf{y} \in \mathbb{R}^{D_L}$ denote the image and the ground-truth label, respectively. Here, $H$, $W$, $D_I$, and $D_L$ denote the image height, image width, number of image channels, and label dimensions, respectively. For each image $\mathbf{x}$, our model $\mathcal{M}$ makes a prediction $\hat{\mathbf{y}} = \mathcal{M}(\mathbf{x}) \in \mathbb{R}^{D_L}$. Given a target image-label pair $(\mathbf{x}^{\text{tgt}}, \mathbf{y}^{\text{tgt}})$ and $\mathbf{y} \neq \mathbf{y}^{\text{tgt}}$, a *traditional attacker* aims to synthesize adversarial examples $\{\mathbf{x}^{\text{adv}}\}$ by adding pixel-wise perturbations to or spatially transforming the original image $\mathbf{x}$ such that $\mathcal{M}(\mathbf{x}^{\text{adv}}) = \mathbf{y}^{\text{tgt}}$.

In this work, we introduce the concept of *semantic attacker* that aims at generating adversarial examples by adding semantically meaningful perturbation with a conditional generative model $\mathcal{G}$. Compared to traditional attacker that usually produces pixel-wise perturbations, the proposed method is able to produce semantically meaningful perturbations.

**Semantic image editing.** For simplicity, we start with the formulation where the input attribute is represented as a compact vector. This formulation can be directly extended to other input attribute formats including semantic layouts. Let $\mathbf{c} \in \mathbb{R}^{D_C}$ be an attribute representation reflecting the semantic factors (e.g., expression or hair color of a portrait image) of image $\mathbf{x}$, where $D_C$ indicates the attribute dimension and $c_i \in \{0, 1\}$ indicates the appearance of $i$-th attribute. Here, our goal is to use the conditional generator for semantic image editing. For example, given a portrait image of a girl with black hair and blonde hair as the new attribute, our generator is supposed to synthesize a new image that turns the girl's hair from black to blonde. More specifically, we denote the new attribute as $\mathbf{c}^{\text{new}} \in \mathbb{R}^{D_C}$ such that the synthesized image is given by $\mathbf{x}^{\text{new}} = \mathcal{G}(\mathbf{x}, \mathbf{c}^{\text{new}})$. In the special case when there is no attribute change ($\mathbf{c} = \mathbf{c}^{\text{new}}$), the generator simply reconstructs the input: $\mathbf{x} = \mathcal{G}(\mathbf{x}, \mathbf{c})$. Supported by the findings mentioned in (Bengio et al., 2013; Reed et al., 2014), our synthesized image $\mathbf{x}^{\text{new}}$ should fall close to the data manifold if we constrain the change of attribute values to be sufficiently small (e.g., we only update one semantic attribute at a time). In addition, we can potentially generate many such images by linearly interpolating between the semantic embeddings of the conditional generator $\mathcal{G}$ using original image $\mathbf{x}$ and the synthesized image $\mathbf{x}^{\text{new}}$ with the augmented attribute.

**Attribute-space interpolation.** We start with a simple solution (detailed in Eq. 1) assuming the adversarial example can be found by directly interpolating in the attribute-space. Given a pair of attributes $\mathbf{c}$ and $\mathbf{c}^{\text{new}}$, we introduce an interpolation parameter $\alpha \in (0, 1)$ to generate the augmented attribute vector $\mathbf{c}^* \in \mathbb{R}^{D_C}$ (see Eq. 1). Given augmented attribute $\mathbf{c}^*$ and original image $\mathbf{x}$, we produce the synthesized image by the generator $\mathcal{G}$. For our notation purpose, we also introduce a delegated function $\mathcal{T}_{\mathcal{G}}$ as a re-parametrization for the generator $\mathcal{G}$. Our formulation is also supported by the empirical results on attribute-conditioned image progression (Yan et al., 2016; Radford et al., 2015) that a well-trained generative model has the capability to synthesize a sequence of images with smooth attribute transitions.

$$
\begin{aligned}
\mathbf{x}^{\text{adv}} &= \operatorname{argmin}_{\alpha} \mathcal{L}(\mathcal{T}_{\mathcal{G}}(\alpha; \mathbf{x}, \mathbf{c}, \mathbf{c}^{\text{new}})) \\
&\text{where } \mathcal{T}_{\mathcal{G}}(\alpha; \mathbf{x}, \mathbf{c}, \mathbf{c}^{\text{new}}) = \mathcal{G}(\mathbf{x}, \mathbf{c}^*) \text{ and } \mathbf{c}^* = \alpha \cdot \mathbf{c} + (1 - \alpha) \cdot \mathbf{c}^{\text{new}}
\end{aligned} \tag{1}
$$

**Feature-map interpolation.** Alternatively, we propose to interpolate using the feature map produced by the generator $\mathcal{G} = \mathcal{G}_{\text{dec}} \circ \mathcal{G}_{\text{enc}}$. Here, $\mathcal{G}_{\text{enc}}$ is the encoder module that takes the image as input and outputs the feature map. Similarly, $\mathcal{G}_{\text{dec}}$ is the decoder module that takes the feature map as input and outputs the synthesized image. Let $\mathbf{f}^* = \mathcal{G}_{\text{enc}}(\mathbf{x}, \mathbf{c}) \in \mathbb{R}^{H_F \times W_F \times C_F}$ be the feature map of an intermediate layer in the generator, where $H_F$, $W_F$ and $C_F$ indicate the height, width, and

number of channels in the feature map.

$$\mathbf{x}^{\mathrm{adv}} = \mathrm{argmin}_{\alpha} \mathcal{L}(\mathcal{T}_{\mathcal{G}}(\alpha; \mathbf{x}, \mathbf{c}, \mathbf{c}^{\mathrm{new}}))$$
$$\text{where } \mathcal{T}_{\mathcal{G}}(\alpha; \mathbf{x}, \mathbf{c}, \mathbf{c}^{\mathrm{new}}) = \mathcal{G}_{\mathrm{dec}}(\mathbf{f}^{*}),$$
$$\mathbf{f}^{*} = \alpha \odot \mathcal{G}_{\mathrm{enc}}(\mathbf{x}, \mathbf{c}) + (\mathbf{1} - \alpha) \odot \mathcal{G}_{\mathrm{enc}}(\mathbf{x}, \mathbf{c}^{\mathrm{new}}) \quad (2)$$

Compared to attribute-space interpolation which is parameterized by a scalar, we parameterize feature-map interpolation by a tensor $\alpha \in \mathbb{R}^{H_F \times W_F \times C_F}$ ($\alpha_{h,w,k} \in (0,1)$, where $1 \le h \le H_F$, $1 \le w \le W_F$, and $1 \le k \le C_F$) with the same shape as the feature map. Compared to linear interpolation over attribute-space, such design introduces more flexibility when interpolating between the original image and the synthesized image. Empirical results in Section 4.2 show our design is critical to the adversarial attack success rate.

## 3.2 ADVERSARIAL OPTIMIZATION OBJECTIVES

As we see in Eq. 3, we obtain the adversarial image $\mathbf{x}^{\mathrm{adv}}$ by minimizing the objective $\mathcal{L}(\cdot)$ with respect to the synthesized image $\mathcal{T}_{\mathcal{G}}(\alpha; \mathbf{x}, \mathbf{c}, \mathbf{c}^{\mathrm{new}})$, which is defined in Eq.(1) and Eq.(2) respectively. Here, each synthesized image $\mathcal{T}_{\mathcal{G}}(\alpha; \mathbf{x}, \mathbf{c}, \mathbf{c}^{\mathrm{new}})$ is produced by the interpolation using the conditional generator $\mathcal{G}$. In our objective function, the first term is the adversarial metric, the second term is a smoothness constraint, and $\lambda$ is used to control the balance between the two terms. The adversarial metric is minimized once the model $\mathcal{M}$ has been successfully attacked towards the target image-label pair $(\mathbf{x}^{\mathrm{tgt}}, \mathbf{y}^{\mathrm{tgt}})$. For identify verification, $\mathbf{y}^{\mathrm{tgt}}$ is the identity representation of the target image; For structured prediction tasks in our paper, $\mathbf{y}^{\mathrm{tgt}}$ either represents certain coordinates (landmark detection) or semantic label maps (semantic segmentation).

$$\mathbf{x}^{\mathrm{adv}} = \mathrm{argmin}_{\alpha} \mathcal{L}(\mathcal{T}_{\mathcal{G}}(\alpha; \mathbf{x}, \mathbf{c}, \mathbf{c}^{\mathrm{new}}))$$
$$\mathcal{L}(\mathcal{T}_{\mathcal{G}}(\alpha; \mathbf{x}, \mathbf{c}, \mathbf{c}^{\mathrm{new}})) = \mathcal{L}_{\mathrm{adv}}(\mathcal{T}_{\mathcal{G}}(\alpha; \mathbf{x}, \mathbf{c}, \mathbf{c}^{\mathrm{new}}); \mathcal{M}, \mathbf{y}^{\mathrm{tgt}}) + \lambda \cdot \mathcal{L}_{\mathrm{smooth}}(\alpha) \quad (3)$$

**Identity verification.** In the identity verification task, two images are considered to be the same identity if the corresponding identity embeddings from the verification model $\mathcal{M}$ are reasonably close.

$$\mathcal{L}_{\mathrm{adv}}(\mathcal{T}_{\mathcal{G}}(\alpha; \mathbf{x}, \mathbf{c}, \mathbf{c}^{\mathrm{new}}); \mathcal{M}, \mathbf{y}^{\mathrm{tgt}}) = \max\left(\kappa, \Phi_{\mathcal{M}}^{\mathrm{id}}(\mathcal{T}_{\mathcal{G}}(\alpha; \mathbf{x}, \mathbf{c}, \mathbf{c}^{\mathrm{new}}), \mathbf{x}^{\mathrm{tgt}})\right) \quad (4)$$

As we see in Eq. 4, $\Phi_{\mathcal{M}}^{\mathrm{id}}(\cdot, \cdot)$ measures the distance between two identity embeddings from the model $\mathcal{M}$, where the normalized $L_2$ distance is used in our setting. In addition, we introduce the parameter $\kappa$ representing the constant related to the false positive rate (FPR) threshold computed from the development set.

**Structured prediction.** For structured prediction tasks such as landmark detection and semantic segmentation, we use Houdini objective proposed in Cisse et al. (2017) as our adversarial metric and select the target landmark (semantic segmentation) target as $\mathbf{y}^{\mathrm{tgt}}$. In addition, $\Phi_{\mathcal{M}}(\cdot, \cdot)$ is a scoring function for each image-label pair and $\gamma$ is the threshold.

$$\mathcal{L}_{\mathrm{adv}}(\mathcal{T}_{\mathcal{G}}(\alpha; \mathbf{x}, \mathbf{c}, \mathbf{c}^{\mathrm{new}}); \mathcal{M}, \mathbf{y}^{\mathrm{tgt}}) = P_{\gamma \sim \mathcal{N}(0,1)}[\Phi_{\mathcal{M}}(\mathcal{T}_{\mathcal{G}}(\alpha; \mathbf{x}, \mathbf{c}, \mathbf{c}^{\mathrm{new}}), \mathbf{y}^{*})$$
$$- \Phi_{\mathcal{M}}(\mathcal{T}_{\mathcal{G}}(\alpha; \mathbf{x}, \mathbf{c}, \mathbf{c}^{\mathrm{new}}), \mathbf{y}^{\mathrm{tgt}}) < \gamma] \cdot l(\mathbf{y}^{*}, \mathbf{y}^{\mathrm{tgt}}) \quad (5)$$

where $\mathbf{y}^{*} = \mathcal{M}(\mathcal{T}_{\mathcal{G}}(\alpha; \mathbf{x}, \mathbf{c}, \mathbf{c}^{\mathrm{new}}))$ and $l(\mathbf{y}^{*}, \mathbf{y}^{\mathrm{tgt}})$ is task loss decided by the specific adversarial target.

**Interpolation smoothness $\mathcal{L}_{\mathrm{smooth}}$.** As the tensor to be interpolated in the feature-map space has far more parameters compared to the attribute itself, we propose to enforce a smoothness constraint on the tensor $\alpha$ used in feature-map interpolation. As we see in Eq. 6, the smoothness loss encourages the interpolation tensors to consist of piece-wise constant patches spatially, which has been widely used as a pixel-wise de-noising objective for natural image processing (Mahendran & Vedaldi, 2015; Johnson et al., 2016).

$$\mathcal{L}_{\mathrm{smooth}}(\alpha) = \sum_{h=1}^{H_F - 1} \sum_{w=1}^{W_F} \|\alpha_{h+1,w} - \alpha_{h,w}\|_2^2 + \sum_{h=1}^{H_F} \sum_{w=1}^{W_F - 1} \|\alpha_{h,w+1} - \alpha_{h,w}\|_2^2 \quad (6)$$

## 4 EXPERIMENTS

In the experimental section, we mainly focus on analyzing the proposed *SemanticAdv* in attacking state-of-the-art face recognition systems on CelebA (Liu et al., 2015) due to its wide applicability (e.g., identification for mobile payment) in the real world. In addition, we extend our attack to urban street scenes with semantic label maps as the condition. We attack the semantic segmentation model DRN-D-22 (Yu et al., 2017) previously trained on Cityscape (Cordts et al., 2016) by generating adversarial examples with dynamic objects manipulated (e.g., insert a car into the scene).

The experimental section is organized as follows. First, we analyze the quality of generated adversarial examples and qualitatively compare our method with $\mathcal{L}_p$ bounded pixel-wise optimization-based method (Carlini & Wagner, 2017; Dong et al., 2018; Xie et al., 2019). Second, we provide both qualitative and quantitative results by controlling each of the semantic attributes at a time. In terms of attack transferability, we evaluate our proposed *SemanticAdv* on various settings and further demonstrate the effectiveness of our method via query-free black-box attacks against online face verification platforms. Third, we compare our method with the baseline against different defense methods on the face verification task. Fourth, we demonstrate that the proposed *SemanticAdv* also applies to the face landmark detection and street-view semantic segmentation.

### 4.1 EXPERIMENTAL SETUP

**Face identity verification.** We select `ResNet-50` and `ResNet-101` (He et al., 2016) trained on MS-Celeb-1M (Guo et al., 2016) as our face verification models. The models are trained using two different objectives, namely, `softmax` loss (Sun et al., 2014; Zhang et al., 2018) and `cosine` loss (Wang et al., 2018a). For simplicity, we use the notation "R-N-S" to indicate the model with $N$-layer residual blocks as backbone trained using `softmax` loss, while "R-N-C" indicates the same backbone trained using `cosine` loss. The distance between face features is measured by normalized L2 distance. For R-101-S model, we decide the parameter $\kappa$ based on the false positive rate (FPR) for the identity verification task. Three different FPRs have been used: $10^{-3}$ (with $\kappa = 1.24$), $3 \times 10^{-4}$ (with $\kappa = 1.05$), and $10^{-4}$ (with $\kappa = 0.60$). The distance metrics and selected thresholds are commonly used when evaluating the performance of face recognition model Klare et al. (2015); Kemelmacher-Shlizerman et al. (2016). Please check the Appendix (see Table B) for more details. To distinguish between the FPR we used in generating adversarial examples and the other FPR used in evaluation, we introduce two notations "Generating FPR (G-FPR)" and "Test FPR (T-FPR)". For the experiment with query-free black-box API attacks, we use the online face verification services provided by Face++ (fac) and AliYun (ali).

**Face landmark detection.** We select Face Alignment Network (FAN) (Bulat & Tzimiropoulos, 2017b) trained on 300W-LP (Zhu et al., 2016b) and fine-tuned on 300-W (Sagonas et al., 2013) for 2D landmark detection. The network is constructed by stacking Hour-Glass network (Newell et al., 2016) with hierarchical block (Bulat & Tzimiropoulos, 2017a). Given a portrait image as input, FAN outputs 2D heatmaps which can be subsequently leveraged to yield 68 2D landmarks.

**Semantic attacks on face images.** In our experiments, we randomly sample $1,280$ distinct identities form CelebA (Liu et al., 2015). To reduce the reconstruction error brought by the generator (e.g., $\mathbf{x} \neq \mathcal{G}(\mathbf{x}, \mathbf{c})$) in practice, we take one more step to obtain the updated feature map $\mathbf{f}' = \mathcal{G}_{\text{enc}}(\mathbf{x}', \mathbf{c})$, where $\mathbf{x}' = \operatorname{argmin}_{\mathbf{x}'} \|\mathcal{G}(\mathbf{x}', \mathbf{c}) - \mathbf{x}\|$ in feature-map interpolation. In our experiments, we use the last `conv` layer before `upsampling` in the generator as our as feature-map $\mathbf{f}$ given by the attack effectiveness. We also fix the parameter $\lambda$ (e.g., balances the adversarial loss and smoothness constraint in Eq. 3) to be $0.01$ for both face verification and landmark detection.

We used the StarGAN (Choi et al., 2018) for attribute-conditional image editing. In particular, we re-trained model on CelebA dataset (Liu et al., 2015) by aligning the face landmarks and then resizing images to resolution $112 \times 112$. In addition, we select 17 identity-preserving attributes as our input condition, as such attributes related to facial expression and hair color.

For each distinct identity pair $(\mathbf{x}, \mathbf{x}^{\text{tgt}})$, we perform *semanticAdv* guided by each of the 17 attributes (e.g., we intentionally add or remove one specific attribute while keeping the rest unchanged). In total, for each image $\mathbf{x}$, we generate 17 adversarial images with different augmented attributes. In the experiments, we select a pixel-wise adversarial attack method (Carlini & Wagner, 2017) (referred as CW) as our baseline for comparison. Compared to our proposed method, CW does not require visual attributes as part of the system, as it only generates one adversarial example for each instance. We refer the corresponding attack success rate as the instance-wise success rate in which the attack

success rate is calculated for each instance. For each instance with 17 adversarial images using different augmented attributes, if one of the 17 resulting images can attack successfully, we count the attack of this instance as one success, vice verse.

**Semantic attacks on street-view images.** We select DRN-D-22 (Yu et al., 2017) as our semantic segmentation model and fine-tune the model on image regions with resolution $256 \times 256$. To synthesize semantic adversarial perturbations, we consider semantic label maps as the input attribute and leverage a generative image manipulation model (Hong et al., 2018) pre-trained on CityScape (Cordts et al., 2016) dataset. Given an input semantic label map at resolution $256 \times 256$, we select a target object instance (e.g., a pedestrian) to attack. Then, we create a manipulated semantic label map by inserting another object instance (e.g., a car) in the vicinity of the target object. Similar to the experiments in the face domain, for both semantic label maps, we used the image manipulation encoder to extract features (with $1,024$ channels at spatial resolution $16 \times 16$) and conducted feature-space interpolation. We synthesized the final image by feeding the interpolated features to the image manipulation decoder. By searching the interpolation coefficient that maximizes the attack rate, we are able to fool the segmentation model with the synthesized final image.

### 4.2 *SemanticAdv* ON FACE IDENTITY VERIFICATION

**Attribute-space vs. feature-space interpolation.** First, we found that both attribute-space and feature-space interpolation could generate reasonable samples (see Figure I in Appendix). Compared to attribute-space interpolation, generating adversarial examples with feature-space interpolation produced much better quantitative results (see Table E in Appendix). We measured the attack success rate of attribute-space interpolation (with G-FPR = T-FPR = $10^{-3}$): $0.08\%$ on R-101-S, $0.31\%$ on R-101-C, and $0.16\%$ on both R-50-S and R-50-C. While feature-space interpolation achieves almost 100% success rate on all those models (see Figure 3). We conjecture that this is because the high dimensional feature space can provide more manipulation freedom.

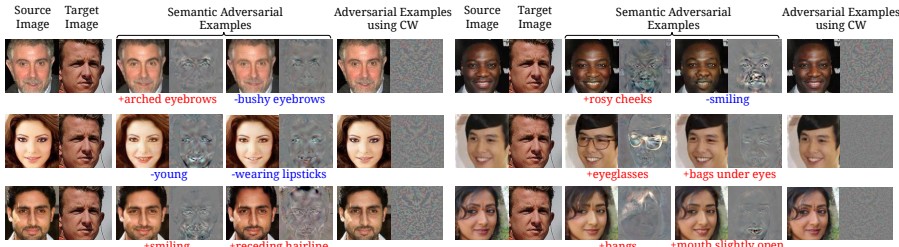

Figure 2: Qualitative comparisons between our proposed *SemanticAdv* and pixel-wise adversarial examples generated by CW (Carlini & Wagner, 2017). Along with the adversarial examples, we also provide the corresponding perturbations (residual) on the right. Perturbations generated by our *SemanticAdv* (G-FPR = $10^{-3}$) are unrestricted with semantically meaningful patterns. More results are shown in Appendix (see Figure N).

**Overall analysis.** Figure 2 shows the generated adversarial images and corresponding perturbations against R-101-S of *SemanticAdv* and CW respectively. The text below each figure is the name of augmented attribute, the sign before the name represents "adding" (in red) or "removing" (in blue) the corresponding attribute from the original image. We see that *SemanticAdv* is able to generate perceptually reasonable examples guided by the corresponding attribute. In particular, *SemanticAdv* is able to generate perturbations on the corresponding regions correlated with the augmented attribute, while the perturbations of CW have no specific pattern and are evenly distributed across the image.

**Analysis: controlling single attribute.** One of the key advantages of *SemanticAdv* is that we can generate adversarial perturbations in a more controllable fashion guided by the semantic attributes. This allows analyzing the robustness of a recognition system against different types of semantic attacks. We group the adversarial examples by augmented attributes in various settings. In Figure 3, we present the attack success rate against two face verification models, namely, R-101-S and R-101-C, guided by different attributes. We highlight the bar with light blue for G-FPR equals to $10^{-3}$ and blue for G-FPR equals to $10^{-4}$, respectively. As we see in this figure, with a larger T-FPR $10^{-3}$, our *SemanticAdv* can achieve almost 100% attack success rate across different attributes. With a smaller T-FPR $10^{-4}$, we find that *SemanticAdv* guided by some attributes such as Mouth Slightly Open and Arched Eyebrows achieve less than 50% attack success rate, while the other attributes

such as `Pale Skin` and `Eyeglasses` are relatively less affected. In summary, we found that *SemanticAdv* guided by attributes describing the local shape (e.g., mouth, earrings) achieve a relatively lower attack success rate compared to attributes relevant to the color (e.g., hair color) or entire face region (e.g., skin). This suggests that the face verification models used in our experiments are more robustly trained in terms of detecting local shapes compared to colors. Please note that in practice we have the flexibility to select attributes for attacking an image based on the perceptual quality and attack success rate.

Figure 4 shows the adversarial examples with augmented semantic attributes against R-101-S model. The attribute names are shown in the bottom. The upper images are $\mathcal{G}(\mathbf{x}, \mathbf{c}^{\text{new}})$ generated by StarGAN with augmented attribute $\mathbf{c}^{\text{new}}$ where the lower images are the corresponding adversarial images with the same augmented attribute.

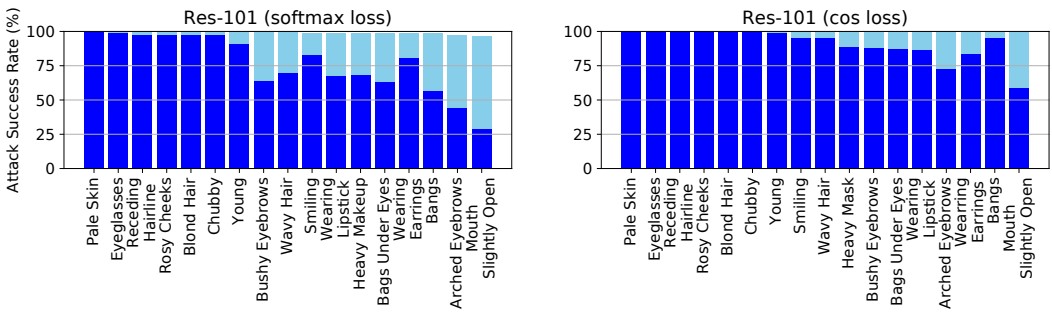

Figure 3: Quantitative analysis on the attack success rate with different single-attribute attacks. In each figure, we show the results correspond to a larger FPR (G-FPR = T-FPR = $10^{-3}$) in skyblue and the results correspond to a smaller FPR (G-FPR = T-FPR = $10^{-4}$) in blue, respectively.

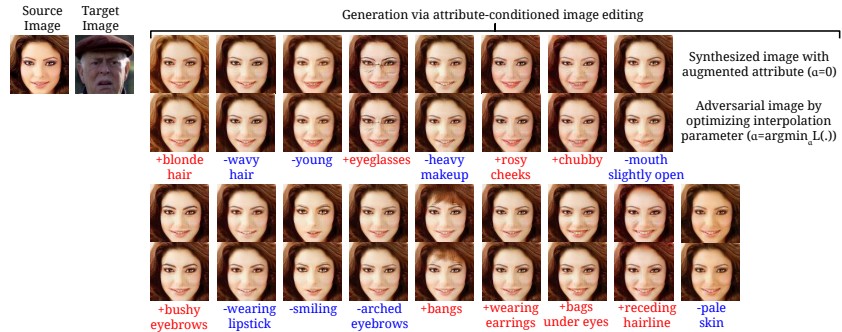

Figure 4: Qualitative analysis on single-attribute adversarial attack (G-FPR = $10^{-3}$). More results are shown in Appendix (see Figure K, Figure L and Figure M).

**Analysis: semantic attack transferability.** To further understand the property of *SemanticAdv*, we analyze the transferability of *SemanticAdv* on various settings. For each model with different FPRs, we select the successfully attacked adversarial examples from Section 4.1 to construct our evaluation dataset and evaluate these adversarial samples across different models. Table 1a illustrates the transferability of *SemanticAdv* among different models by using the same FPRs (G-FPR = T-FPR = $10^{-3}$). Table 1b illustrates the result with different FPRs (G-FPR = $10^{-4}$ and T-FPR = $10^{-3}$) for generation and evaluation. As shown in Table 1a, adversarial examples generated against models trained with softmax loss exhibit certain transferability compared to models trained with cosine loss. We conduct the same experiment by generating adversarial examples with CW and found it has weaker transferability compared to our *SemanticAdv* (results in brackets of Table 1).

As Table 1b illustrates, the adversarial examples generated against the model with smaller G-FPR = $10^{-4}$ exhibit strong attack success rate when evaluating on the model with larger T-FPR = $10^{-3}$. Especially, we found the adversarial examples generated against R-101-S have the best attack performance on other models. These findings motivate the analysis of query-free black-box API attack detailed in the following paragraph.

| $\mathcal{M}_{\text{test}}$ / $\mathcal{M}_{\text{opt}}$ | R-50-S | R-101-S | R-50-C | R-101-C |
|---|---|---|---|---|
| R-50-S | 1.000 (1.000) | **0.108** (0.007) | **0.023** (0.002) | **0.018** (0.002) |
| R-101-S | **0.169** (0.006) | 1.000 (1.000) | **0.030** (0.002) | **0.032** (0.003) |
| R-50-C | **0.166** (0.019) | **0.202** (0.025) | 1.000 (1.000) | **0.048** (0.007) |
| R-101-C | **0.120** (0.015) | **0.236** (0.029) | **0.040** (0.006) | 1.000 (1.000) |

(a)

| $\mathcal{M}_{\text{test}}$ / $\mathcal{M}_{\text{opt}}$ | R-50-S | R-101-S |
|---|---|---|
| R-50-S | 1.000 (1.000) | **0.862** (0.530) |
| R-101-S | **0.874** (0.422) | 1.000 (1.000) |
| R-50-C | **0.693** (0.347) | **0.837** (0.579) |
| R-101-C | **0.617** (0.218) | **0.888** (0.617) |

(b)

Table 1: Transferability of *SemanticAdv*: cell $(i, j)$ shows attack success rate of adversarial examples generated against $j$-th model and evaluate on $i$-th model. Results of CW are listed in brackets. Left: Results generated with G-FPR = $10^{-3}$ and T-FPR = $10^{-3}$; Right: Results generated with G-FPR = $10^{-4}$ and T-FPR = $10^{-3}$.

**Query-free black-box API attack.** In this experiment, we generate adversarial examples against R-101-S with G-FPR = $10^{-3}(\kappa = 1.24)$, G-FPR = $10^{-4}(\kappa = 0.60)$, and G-FPR $< 10^{-4}(\kappa = 0.20)$, respectively. We evaluate our algorithm on two industry level APIs, namely, Face++ and AliYun face verification platform. Since attack transferability has never been explored in concurrent work that generates semantic adversarial examples, we use $\mathcal{L}_p$ bounded pixel-wise methods (Carlini & Wagner, 2017; Dong et al., 2018; Xie et al., 2019) as our baselines. As we see Table 2, which shows the best performance of each method, our *SemanticAdv* achieves much higher attack success rate than CW in both APIs with all FPR thresholds (e.g., our adversarial examples generated with G-FPR $< 10^{-4}$ achieves $64.63\%$ attack success rate on Face++ platform with T-FPR = $10^{-3}$). In addition, we found that lower G-FPR can achieve higher attack success rate on APIs within the same T-FPR (see Table I in Appendix).

| API name | Face++ | | AliYun | |
|---|---|---|---|---|
| Attacker / Evaluation Metric | T-FPR = $10^{-3}$ | T-FPR = $10^{-4}$ | T-FPR = $10^{-3}$ | T-FPR = $10^{-4}$ |
| Dong et al. (2018) | 30.77 | 21.03 | 18.00 | 6.50 |
| Xie et al. (2019) | 37.95 | 25.64 | 21.50 | 11.00 |
| CW (G-FPR $< 10^{-4}$) | 41.62 | 24.37 | 19.00 | 12.00 |
| *SemanticAdv* (G-FPR $< 10^{-4}$) | **64.63** | **42.69** | **35.50** | **22.17** |

Table 2: Quantitative analysis on query-free black-box attack. We use ResNet-101 optimized with `softmax` loss for evaluation and report the attack success rate(%) on two online face verification platforms. Note that for PGD-based attacks, we adopt MI-FGSM ($\epsilon = 8$) in Dong et al. (2018) and M-DI$^2$-FGSM ($\epsilon = 8$) in Xie et al. (2019), respectively.

**User study.** To measure the perceptual quality of the adversarial images generated by *SemanticAdv*, we conduct a user study on Amazon Mechanical Turk (AMT). We use the adversarial examples generated with G-FPR $< 10^{-4}$, which is the most strict setting in our experiment, to conduct the user study for both CW and *SemanticAdv*. In total, we collect $2,620$ annotations from 77 participants. In $39.14\pm1.96\%$ (close to random guess $50\%$) of trials the adversarial images generated by *SemanticAdv* are selected as reasonably-looking images and in $30.27 \pm 1.96\%$ of trials, the adversarial images generated by CW are selected as reasonably-looking. It indicates that *SemanticAdv* can generate more reasonable-looking adversarial examples compared with CW under the most strict setting with G-FPR $< 10^{-4}$. Qualitative comparisons are shown in Appendix (see Figure H).

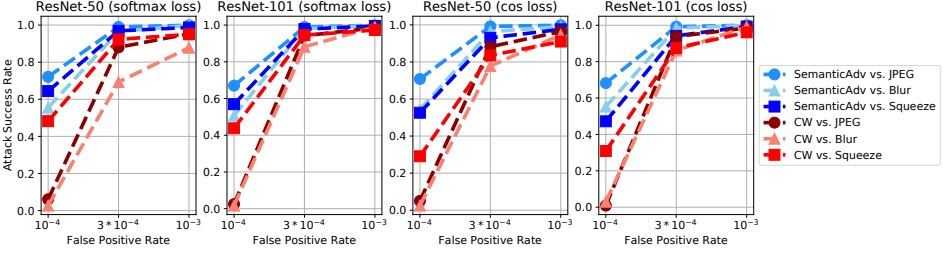

Figure 5: Quantitative analysis on attacking several defense methods including `JPEG` (Dziugaite et al., 2016), `Blurring` (Li & Li, 2017), and `Feature Squeezing` (Xu et al., 2017).

***SemanticAdv* against defense methods.** We evaluate the strength of the proposed attack by testing against four existing defense methods, namely, `Feature squeezing` (Xu et al., 2017), `Blurring` (Li & Li, 2017), `JPEG` (Dziugaite et al., 2016) and `AMI` (Tao et al., 2018). For `AMI` (Tao et al., 2018), we first extract attribute witnesses with our aligned face images and then leverage them

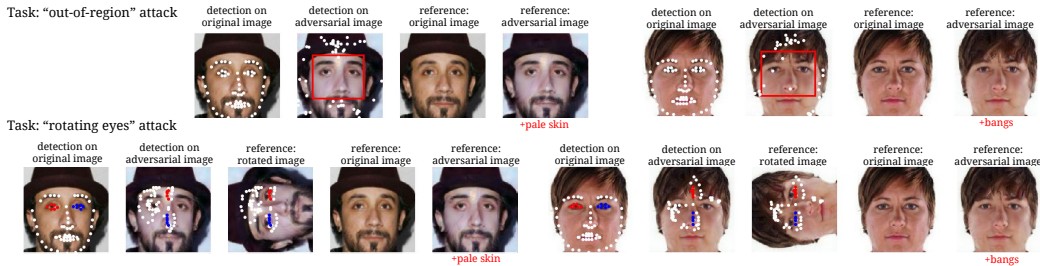

Figure 6: Qualitative results on attacking face landmark detection model.

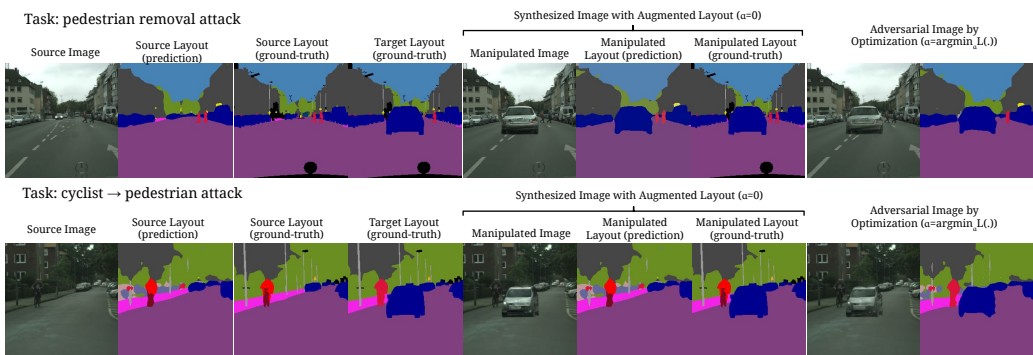

Figure 7: Qualitative results on attacking street-view semantic segmentation model.

to construct attribute-steered model. We use fc7 of pretrained VGG (Parkhi et al., 2015) as the face representation. AMI yields a consistency score for each face image to indicate whether it is a benign image. The score is measured by the cosine similarity between the representations from original model and attribute-steered model. With $10\%$ false positives on benign inputs, it only achieves $8\%$ detection accuracy for *SemanticAdv* and $12\%$ detection accuracy for CW.

Figure 5 illustrates *SemanticAdv* is more robust against these defense methods comparing with CW. The same G-FPR and T-FPR are used for evaluation. Under the condition that T-FPR is $10^{-3}$, both *SemanticAdv* and CW achieve high attack success rate, while *SemanticAdv* marginally outperforms CW when FPR goes down to $10^{-4}$. While defense methods have proven to be effective against CW attacks on classifiers trained with ImageNet (Krizhevsky et al., 2012), our results indicate that these methods are still vulnerable in face verification system with small T-FPR.

## 4.3 *SemanticAdv* ON FACE LANDMARK DETECTION

We also evaluate the effectiveness of *SemanticAdv* on face landmark detection. We select two attack tasks, namely, "Rotating Eyes" and "Out of Region". For the "Rotating Eyes" task, we rotate the coordinates of the eyes in the image counter-clockwise by $90°$. For the "Out of Region" task, we set a target bounding box and attempt to push all points out of the box. We summarize the experimental setup and quantitative results in the Appendix (see Table D). As we see in Figure 6, our method is applicable to attack landmark detection models.

## 4.4 *SemanticAdv* ON STREET-VIEW SEMANTIC SEGMENTATION

We further demonstrate the applicability of our *SemanticAdv* beyond face domain by generating adversarial perturbations on street-view images. Figure 7 illustrates the adversarial examples on semantic segmentation. In the first example, we select the leftmost the pedestrian as the target object instance and insert another car into the scene to attack it. The segmentation model has been successfully attacked to neglect the pedestrian (see last column), while it does exist in the scene (see second-to-last column). In the second example, we insert an adversarial car in the scene by *SemanticAdv* and the cyclist has been recognized as a pedestrian by the segmentation model.

## 5 CONCLUSIONS

Overall we presented a novel attack method *SemanticAdv*, which is capable of generating unrestricted adversarial perturbations guided by semantic attributes edition. Compared to existing methods, *SemanticAdv* works in a more controllable fashion. Experimental evaluations on face verification and landmark detection demonstrate several unique properties including attack transferability. We believe this work would open up new research opportunities and challenges in the field of adversarial learning. For instance, how to leverage semantic information to defend against such attacks will lead to potential new discussion.

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

## A  FACE IDENTITY VERIFICATION

**Benchmark performance.**   First, we provide additional information about the ResNet models we used in the experiments. We summarize in Table A the performance on several face identity verification benchmarks including Labeled Face in the Wild (LFW) dataset (Huang et al., 2008), AgeDB-30 dataset (Moschoglou et al., 2017), and Celebrities in Frontal-Profile (CFP) dataset (Sengupta et al., 2016).

| $\mathcal{M}$ / benchmarks | LFW | AgeDB-30 | CFP-FF | CFP-FP |
|:---:|:---:|:---:|:---:|:---:|
| R-50-S | 99.27 | 94.15 | 99.26 | 91.49 |
| R-101-S | 99.42 | 95.93 | 99.57 | 95.07 |
| R-50-C | 99.38 | 95.08 | 99.24 | 90.24 |
| R-101-C | 99.67 | 95.58 | 99.57 | 92.71 |

Table A: The performnace of ResNet models on several benchmark datasets.

**Identity verification thresholds.**   Table B shows the threshold values used in our experiments when determining whether two portrait images belong to the same identity or not. The selected FPR thresholds and normalized L2 distance between face features are commonly used when evaluating the performance of face recognition model Klare et al. (2015); Kemelmacher-Shlizerman et al. (2016).

| FPR/$\mathcal{M}$ | R-50-S | R-101-S | R-50-C | R-101-C |
|:---:|:---:|:---:|:---:|:---:|
| $10^{-3}$ | 1.181 | 1.244 | 1.447 | 1.469 |
| $3 \times 10^{-4}$ | 1.058 | 1.048 | 1.293 | 1.242 |
| $10^{-4}$ | 0.657 | 0.597 | 0.864 | 0.809 |

Table B: The threshold values for face identity verification.

**Implementation details.**   We use Adam (Kingma & Ba, 2015) as the optimizer to generate adversarial examples for both our *SemanticAdv* and CW. More specifically, we run optimization for up to 200 steps with a fixed learning rate 0.05 for cases when G-FPR $\leq 10^{-4}$. Otherwise, we run optimization for up to $500$ steps with a fixed learning rate 0.01. For pixel-wise attack method CW, we use additional pixel reconstruction loss with corresponding loss weight to $5$. We run optimization for up to $1,000$ steps with a fixed learning rate $10^{-3}$.

**Evaluation metrics.**   To evaluate the performance of *semanticAdv* under different attributes, we consider three metrics as follows:

- *Best*: if there is one attribute among 17 attributes that can be successfully attacked, we count the attack success rate for this face identity as 1;

- *Average*: we calculate the average attack success rate among 17 attributes for the same face identity;

- *Worst*: only when all of 17 attributes can be successfully attacked, we count the attack success rate for this person as 1;

Note that, for a fair comparison with CW, we should use the *Best* metric for our *SemanticAdv*, as CW is the traditional pixel-wise attack method works regardless of the attribute. In addition, we report the performance using the *average* and *worst* metric, which actually provides additional insights into the robustness of face verification models across different attributes. For instance, combining the results from Table C and Figure 3, we understand that the face verification models used in our experiments have different levels of robustness across attributes. For example, face verification models are more robust against local shape variations than color variations, e.g., pale skin has higher attack success rate than mouth open. We believe these discoveries will help the community further understand the properties of face verification models.

Table C shows the overall performance (accuracy) of face verification model and attack success rate of *SemanticAdv* and CW. As we can see from Table C, although the face model trained with cosine loss achieves higher face recognition performance, it is more vulnerable to adversarial attack compared with the model trained with softmax loss.

| G-FPR | Metrics / $\mathcal{M}$ | R-50-S | R-101-S | R-50-C | R-101-C |
|---|---|---|---|---|---|
| $10^{-3}$ | Verification Accuracy | 98.36 | 98.78 | 98.63 | 98.84 |
| | $\mathbf{x}'$ | 0.00 | 0.00 | 0.08 | 0.00 |
| | $G(\mathbf{x}', \mathbf{c})$ | 0.00 | 0.00 | 0.00 | 0.23 |
| | $G(\mathbf{x}', \mathbf{c}^{\text{new}})$(*Best*) | 0.16 | 0.08 | 0.16 | 0.31 |
| | *SemanticAdv* (*Best*) | 100.00 | 100.00 | 100.00 | 100.00 |
| | *SemanticAdv* (*Worst*) | 91.95 | 93.98 | 99.53 | 99.77 |
| | *SemanticAdv* (*Average*) | 98.98 | 99.29 | 99.97 | 99.99 |
| | CW | 100.00 | 100.00 | 100.00 | 100.00 |
| $3 \times 10^{-4}$ | Verification Accuracy | 97.73 | 97.97 | 97.91 | 97.85 |
| | $\mathbf{x}'$ | 0.00 | 0.00 | 0.00 | 0.00 |
| | $G(\mathbf{x}', \mathbf{c})$ | 0.00 | 0.00 | 0.00 | 0.00 |
| | $G(\mathbf{x}', \mathbf{c}^{\text{new}})$(*Best*) | 0.00 | 0.00 | 0.00 | 0.00 |
| | *SemanticAdv* (*Best*) | 100.00 | 100.00 | 100.00 | 100.00 |
| | *SemanticAdv* (*Worst*) | 83.75 | 79.06 | 98.98 | 96.64 |
| | *SemanticAdv* (*Average*) | 97.72 | 97.35 | 99.92 | 99.72 |
| | CW | 100.00 | 100.00 | 100.00 | 100.00 |
| $10^{-4}$ | Verification Accuracy | 93.25 | 92.80 | 93.43 | 92.98 |
| | $\mathbf{x}'$ | 0.00 | 0.00 | 0.00 | 0.00 |
| | $G(\mathbf{x}', \mathbf{c})$ | 0.00 | 0.00 | 0.00 | 0.00 |
| | $G(\mathbf{x}', \mathbf{c}^{\text{new}})$(*Best*) | 0.00 | 0.00 | 0.00 | 0.00 |
| | *SemanticAdv* (*Best*) | 100.00 | 100.00 | 100.00 | 100.00 |
| | *SemanticAdv* (*Worst*) | 33.59 | 19.84 | 67.03 | 48.67 |
| | *SemanticAdv* (*Average*) | 83.53 | 76.64 | 95.57 | 91.13 |
| | CW | 100.00 | 100.00 | 100.00 | 100.00 |

Table C: Quantitative result of identity verification (%). It shows accuracy of face verification model and attack success rate of *SemanticAdv* and CW. $\mathbf{x}'$, $G(\mathbf{x}', \mathbf{c})$ and $G(\mathbf{x}', \mathbf{c}^{\text{new}})$ are the intermediate results of our method before adversarial perturbation.

**Defense methods.** `Feature squeezing` (Xu et al., 2017) is a simple but effective method by reducing color bit depth to remove the adversarial effects. We compress the image represented by 8 bits for each channel to 4 bits for each channel to evaluate the effectiveness. For `Blurring` (Li & Li, 2017), we use a $3 \times 3$ Gaussian kernel with standard deviation 1 to smooth the adversarial perturbations. For `JPEG` Dziugaite et al. (2016), it leverages the compression and decompression to remove the adversarial perturbation. We set the compression ratio as 0.75 in our experiment.

# B  FACE LANDMARK DETECTION

**Implementation details.** To perform attack on the face landmark detection model, we run optimization for up to $2,000$ steps with a fixed learning rate $0.05$. We set the balancing factor $\lambda$ (see Eq. 3) to value $0.01$ for this experiment.

**Evaluation Metrics.** We apply two different metrics for two adversarial attack tasks respectively. For "Rotating Eyes" task, we use a well-adopted metric Normalized Mean Error (NME) (Bulat & Tzimiropoulos, 2017b) for experimental evaluation.

$$r_{\text{NME}} = \frac{1}{N} \sum_{k=1}^{N} \frac{||\mathbf{p}_k - \hat{\mathbf{p}}_k||_2}{\sqrt{W_B * H_B}}, \tag{7}$$

where $\mathbf{p}_k$ denotes the $k$-th ground-truth landmark, $\hat{\mathbf{p}}_{\mathbf{k}}$ denotes the $k$-th predicted landmark and $\sqrt{W_B * H_B}$ is the square-root area of ground-truth bounding box, where $W_B$ and $H_B$ represents the width and height of the box.

For "Out of Region" task, we consider the attack is successful if the landmark predictions fall outside a pre-defined centering region on the portrait image. Thus, we introduce a metric that reflects the portion of landmarks outside of the pre-defined centering region: $r_{\text{OUT}} = \frac{N_{\text{out}}}{N_{\text{total}}}$, where $N_{\text{out}}$ denotes the number of predicted landmarks outside the pre-defined bounding box and $N_{\text{total}}$ denotes the total number of landmarks.

| Tasks (Metrics) | Pristine | Augmented Attributes | | | | | | | |
|---|---|---|---|---|---|---|---|---|---|
| | | Blond Hair | Young | Eyeglasses | Rosy Cheeks | Smiling | Arched Eyebrows | Bangs | Pale Skin |
| Rotating eyes ($r_{\text{NME}}$) ↓ | 28.04 | 14.03 | 17.28 | 8.58 | 13.24 | 19.21 | 23.42 | 15.99 | 10.72 |
| Out-of-region ($r_{\text{OUT}}$) ↓ | 45.98 | 17.42 | 23.04 | 7.51 | 16.65 | 25.44 | 33.85 | 20.03 | 13.51 |

Table D: Quantitative results on face landmark detection (%) The two row shows the measured ratios (lower is better) for "Rotating Eyes" and "Out Of Region" task, respectively.

We present the quantitative results of *SemanticAdv* on face landmark detection model in Table D including two adversarial tasks, namely, "Out of Region" and "Rotating Eyes". We observe that our method is efficient to perform attacking on landmark detection models. For certain attributes such as "Eyeglasses", "Plae Skin", *SemanticAdv* is able to achieve reasonably-good performance.

## C  USER STUDY

We conducted a user study on the adversarial images of *SemanticAdv* and CW used in the experiment of API-attack and the original images. The adversarial images are generated with G-FPR $< 10^{-4}$ for both methods. We present a pair of original image and adversarial image to participants and ask them to rank the two options. The order of these two images is randomized and the images are displayed for 2 seconds in the screen during each trial. After the images disappear, the participants have unlimited time to select the more reasonably-looking image according to their perception. For each participant, one could only conduct at most 50 trials, and each adversarial image was shown to 5 different participants. Some qualitative results are shown in Figure H. In total, we collect $2,620$ annotations from 77 participants. In $39.14 \pm 1.96\%$ of trials the adversarial images generated by *SemanticAdv* are selected as reasonably-looking images and in $30.27 \pm 1.96\%$ of trails, the adversarial images generated by CW are selected as reasonably-looking images. It indicates that our semantic adversarial examples are more perceptual reasonably-looking than CW. Additionally, we also conduct the user study with larger G-FPR$= 10^{-3}$. In $45.42 \pm 1.96\%$ of trials, the adversarial images generated by *SemanticAdv* are selected as reasonably-looking images, which is very close to the random guess (50%).

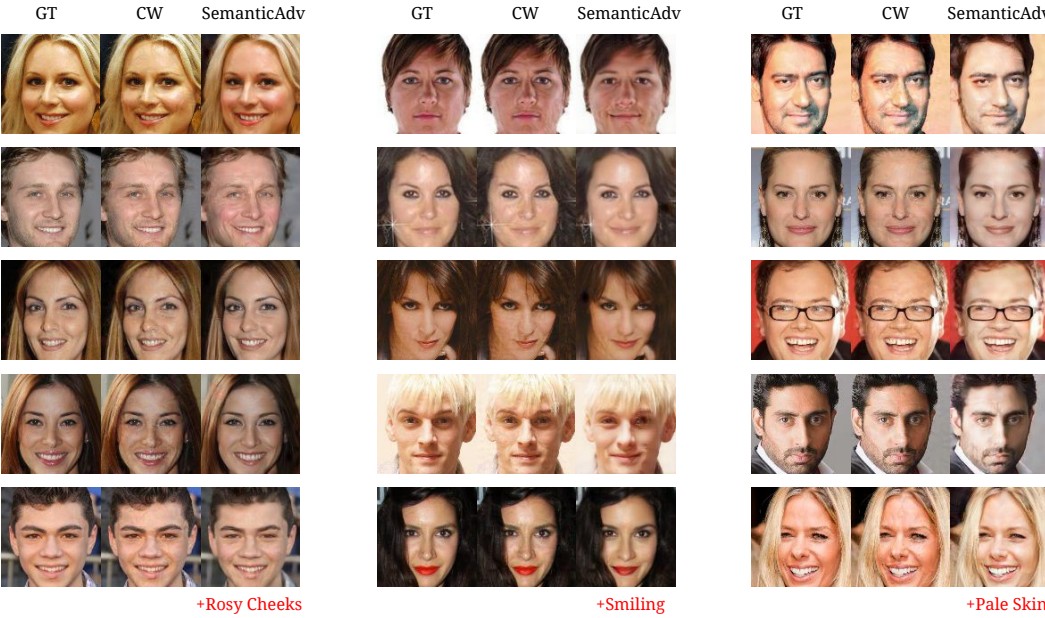

Figure H: Qualitative comparisons among ground truth, pixel-wise adversarial examples generated by CW, and our proposed *SemanticAdv*. Here, we present the results from G-FPR $< 10^{-4}$ so that perturbations are visible.

# D  ABLATION STUDY: FEATURE-SPACE INTERPOLATION

We conduct an ablation study on feature-space interpolation by analyzing attack success rates with different feature-maps in the StarGAN network. Table E shows the attack success rate on R-101-S. Here, we use $\mathbf{f}_i$ to represent the feature-map after $i$-th up-sampling operation. $\mathbf{f}_0$ denotes the feature-map before applying up-sampling operation. The result demonstrates that samples generated by interpolating on $\mathbf{f}_0$ achieve the highest success rate. Since $\mathbf{f}_0$ is the feature-map before decoder, it still well embeds semantic information in the feature space. We adopt $\mathbf{f}_0$ for interpolation in our experiments.

We also conduct a qualitative comparison between attribute-space and feature-space interpolation. As shown in Figure I, images generated by attribute-space and feature-space interpolation are both reasonably-looking.

| T-FPR (G-FPR) | $10^{-3}$ ($10^{-3}$) | | | $3 \times 10^{-4}$ ($3 \times 10^{-4}$) | | | $10^{-4}$ ($10^{-4}$) | | |
|---|---|---|---|---|---|---|---|---|---|
| Layer ($\mathbf{f}$) | $\mathbf{f}_0$ | $\mathbf{f}_1$ | $\mathbf{f}_2$ | $\mathbf{f}_0$ | $\mathbf{f}_1$ | $\mathbf{f}_2$ | $\mathbf{f}_0$ | $\mathbf{f}_1$ | $\mathbf{f}_2$ |
| Attack Success Rate | 99.29 | 98.32 | 75.62 | 97.35 | 94.10 | 57.15 | 76.64 | 67.40 | 19.63 |

Table E: Attack success rate by selecting different layer's feature-map for interpolation on R-101-S(%). $\mathbf{f}_i$ indicates the feature-map after $i$-th up-sampling operation.

| T-FPR (G-FPR) | $10^{-3}$ ($10^{-3}$) | | $3 \times 10^{-4}$ ($3 \times 10^{-4}$) | | $10^{-4}$ ($10^{-4}$) | |
|---|---|---|---|---|---|---|
| Layer ($\mathbf{f}$) | $\mathbf{f}_{-2}$ | $\mathbf{f}_{-1}$ | $\mathbf{f}_{-2}$ | $\mathbf{f}_{-1}$ | $\mathbf{f}_{-2}$ | $\mathbf{f}_{-1}$ |
| Attack Success Rate | 49.40 | 92.09 | 30.44 | 81.87 | 6.66 | 45.46 |

Table F: Attack success rate by selecting different layer's feature-map for interpolation on R-101-S(%). $\mathbf{f}_{-2}$ indicates the feature-map after the last down-sampling operation and $\mathbf{f}_{-2}$ indicates the feature-map after $\mathbf{f}_{-2}$.

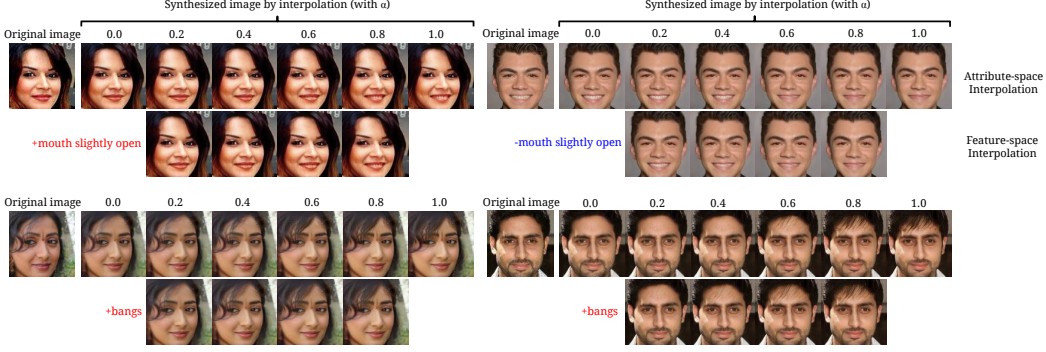

Figure I: Qualitative comparisons between attribute-space and feature-space interpolation.

# E  SEMANTIC ATTACK TRANSFERABILITY

In Table G, we present the quantitative results of the transferability with G-FPR = $10^{-4}$, T-FPR = $10^{-4}$. We observe that with the strict criterion (Lower T-FPR) of the verification model, the transferability becomes lower cross different models.

To explore whether the improvement of transferability in *SemanticAdv* is introduced by semantic editing rather than optimized semantic perturbation in feature space. We add a StrawMan [CW + $\mathbf{x}^{\text{new}}$] baseline which uses a controllable semantic-attribute-based generator to generate semantically different images without any notion of an adversarial attack, and then applies standard $L_p$ CW attacks on that generated image. The results are shown in Table H. The performance of the StrawMan [CW + $\mathbf{x}^{\text{new}}$] is worse than *SemanticAdv*. This result justifies that our *SemanticAdv* is able to produce novel adversarial examples which cannot be simply achieved by combining attribute-conditional image editing model with $L_p$ bounded perturbation.

| $\mathcal{M}_{\text{test}} / \mathcal{M}_{\text{opt}}$ | R-50-S | R-101-S | R-50-C | R-101-C |
|---|---|---|---|---|
| R-50-S | 1.000 | 0.005 | 0.000 | 0.000 |
| R-101-S | 0.000 | 1.000 | 0.000 | 0.000 |
| R-50-C | 0.000 | 0.000 | 1.000 | 0.000 |
| R-101-C | 0.000 | 0.000 | 0.000 | 1.000 |

Table G: Transferability of *SemanticAdv*: cell $(i, j)$ shows attack success rate of adversarial examples generated against $j$-th model and evaluate on $i$-th model. Results generated with G-FPR = $10^{-4}$, T-FPR = $10^{-4}$.

| $\mathcal{M}_{\text{test}} / \mathcal{M}_{\text{opt}}$ | R-101-S |
|---|---|
| R-50-S | 0.035 (0.108) |
| R-101-S | 1.000 (1.000) |
| R-50-C | 0.145 (0.202) |
| R-101-C | 0.085 (0.236) |

(a) G-FPR=$10^{-3}$, T-FPR=$10^{-3}$

| $\mathcal{M}_{\text{test}} / \mathcal{M}_{\text{opt}}$ | R-101-S |
|---|---|
| R-50-S | 0.615 (0.862) |
| R-101-S | 1.000 (1.000) |
| R-50-C | 0.570 (0.837) |
| R-101-C | 0.695 (0.888) |

(b) G-FPR=$10^{-4}$, T-FPR=$10^{-3}$

Table H: Transferability of StrawMan: cell $(i, j)$ shows attack success rate of adversarial examples generated against $j$-th model and evaluate on $i$-th model. Results of *SemanticAdv* are listed in brackets.

## F  QUERY-FREE BLACK-BOX API ATTACK

| API name | Face++ | | AliYun | |
|---|---|---|---|---|
| Attacker / Evaluation Metric | T-FPR = $10^{-3}$ | T-FPR = $10^{-4}$ | T-FPR = $10^{-3}$ | T-FPR = $10^{-4}$ |
| Original $\mathbf{x}$ | 2.04 | 0.51 | 0.50 | 0.00 |
| Generated $\mathbf{x}^{\text{new}}$ | 4.21 | 0.53 | 0.50 | 0.00 |
| CW (G-FPR = $10^{-3}$) | 16.24 | 3.55 | 4.50 | 0.00 |
| StrawMan [CW + $\mathbf{x}^{\text{new}}$] (G-FPR = $10^{-3}$) | 10.71 | 4.08 | 3.00 | 0.00 |
| *SemanticAdv* (G-FPR = $10^{-3}$) | **27.32** | **9.79** | **7.50** | **2.00** |
| CW (G-FPR = $10^{-4}$) | 30.61 | 15.82 | 12.50 | 4.50 |
| StrawMan [CW + $\mathbf{x}^{\text{new}}$] (G-FPR = $10^{-4}$) | 21.32 | 9.14 | 7.50 | 1.50 |
| *SemanticAdv* (G-FPR = $10^{-4}$) | **57.22** | **38.66** | **29.50** | **17.50** |
| Dong et al. (2018) | 30.77 | 21.03 | 18.00 | 6.50 |
| Xie et al. (2019) | 37.95 | 25.64 | 21.50 | 11.00 |
| CW (G-FPR < $10^{-4}$) | 41.62 | 24.37 | 19.00 | 12.00 |
| StrawMan [CW + $\mathbf{x}^{\text{new}}$] (G-FPR < $10^{-4}$) | 27.69 | 15.38 | 10.00 | 3.00 |
| *SemanticAdv* (G-FPR < $10^{-4}$) | **64.63** | **42.69** | **35.50** | **22.17** |

Table I: Quantitative analysis on query-free black-box attack. We use ResNet-101 optimized with `softmax` loss for evaluation and report the attack success rate(%).

In Table I, we present the results of *SemanticAdv* performing query-free black-box attack on two online face verification platforms. *SemanticAdv* outperforms CW and StawnMan in both APIs under all FPR thresholds. In addition, we achieve higher attack success rate on APIs using samples generated using lower G-FPR compared to samples generated using higher G-FPR with the same T-FPR. Original $\mathbf{x}$ and generated $\mathbf{x}^{\text{new}}$ are regarded as reference point of the performance of online face verification platforms.

Figure J illustrates our *SemanticAdv* on attacking Microsoft Azure face verification system, which further demonstrate the effectiveness of our approach.

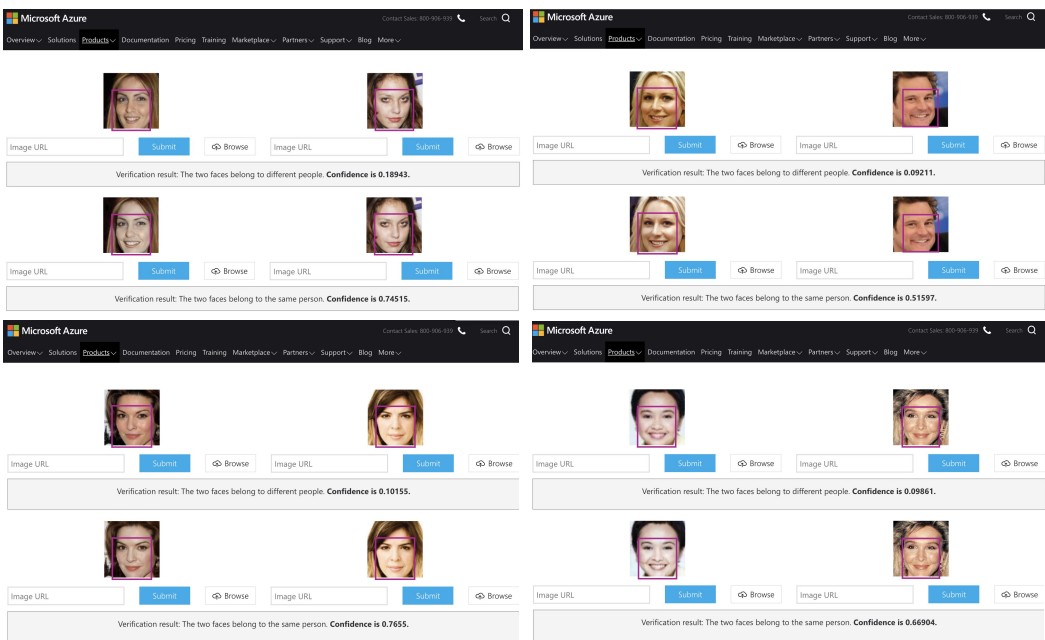

Figure J: Illustration of our *SemanticAdv* in the real world face verification platform (editing on pale skin). Note that the confidence denotes the likelihood that two faces belong to the same person.

# G  MORE VISUALIZATIONS

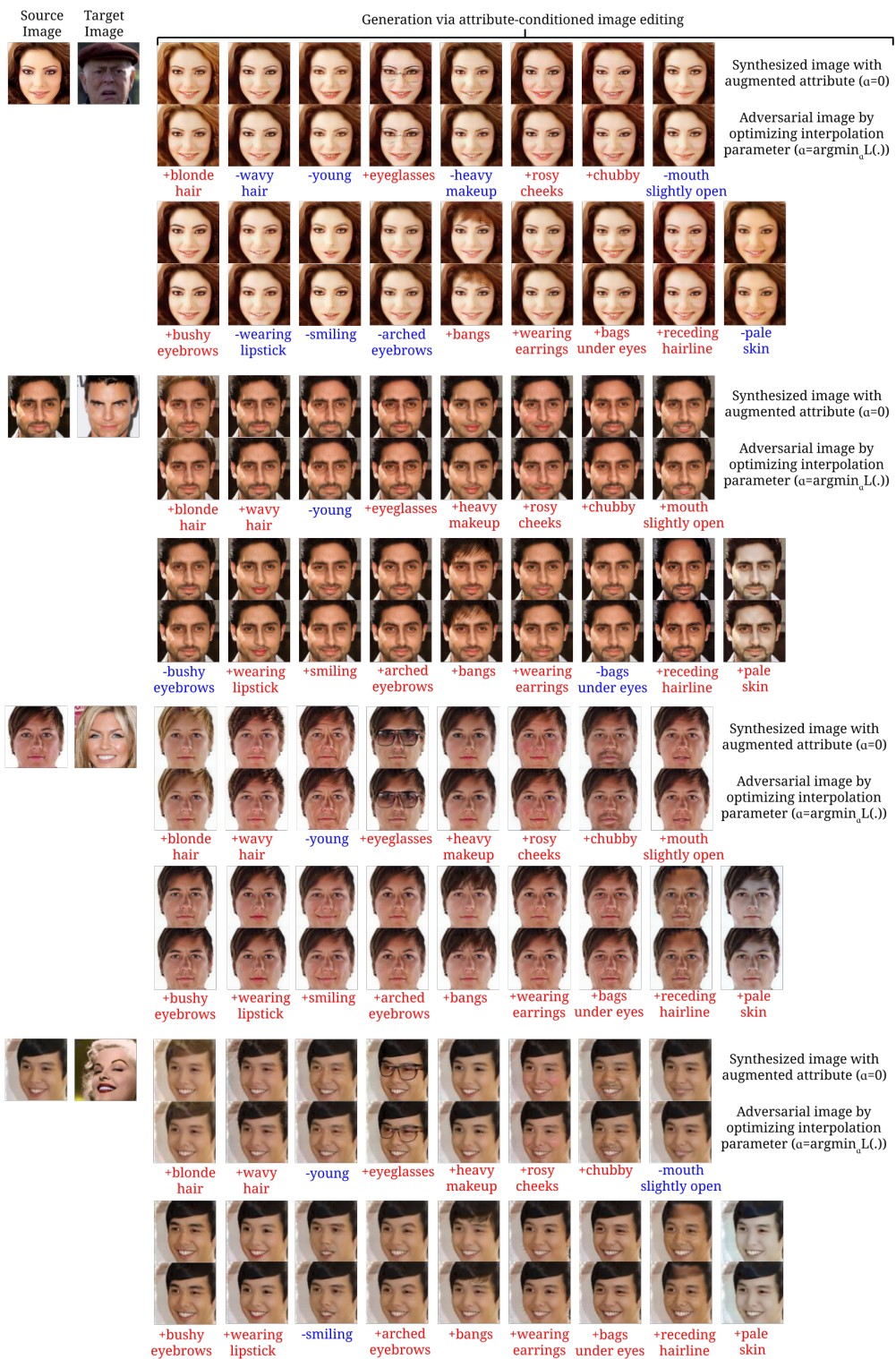

Figure K: Qualitative analysis on single-attribute adversarial attack (G-FPR = $10^{-3}$).

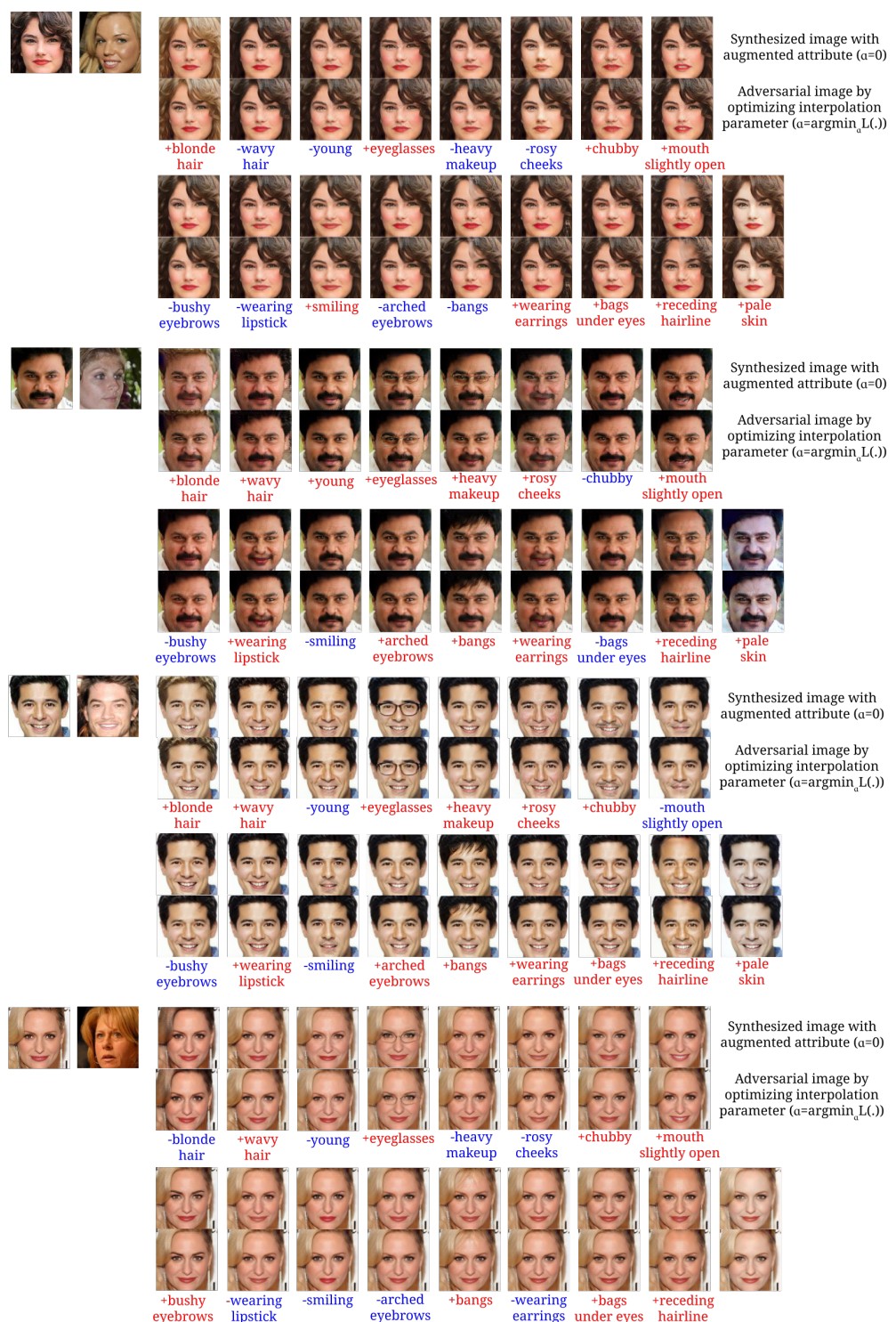

Figure L: Qualitative analysis on single-attribute adversarial attack (G-FPR = $10^{-3}$).

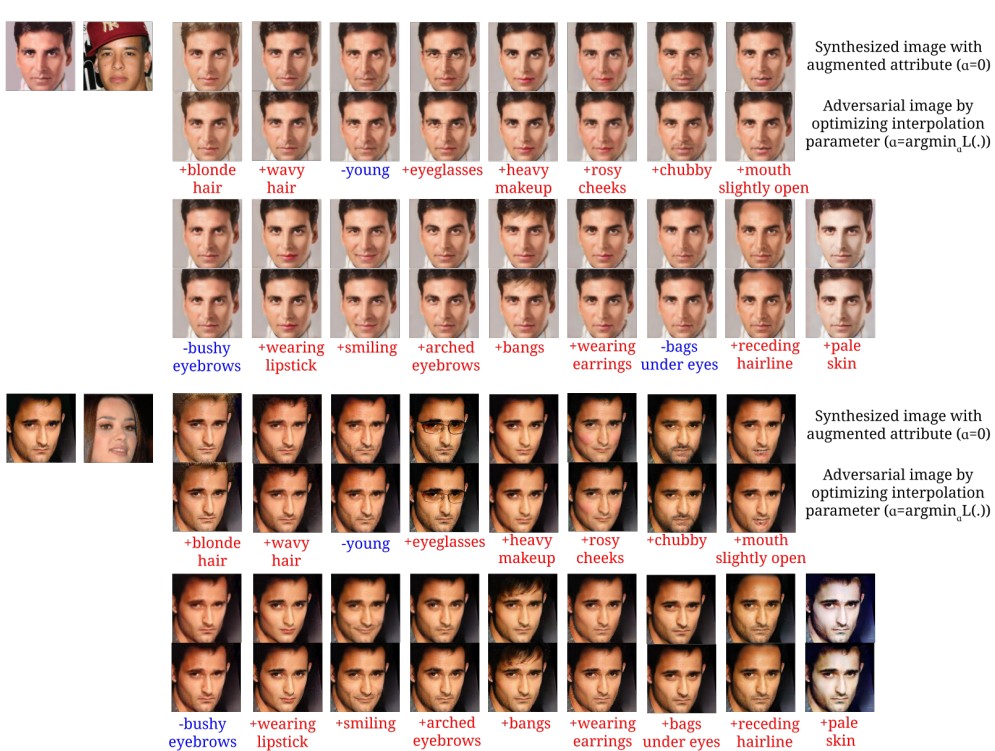

Figure M: Qualitative analysis on single-attribute adversarial attack (G-FPR = $10^{-3}$).

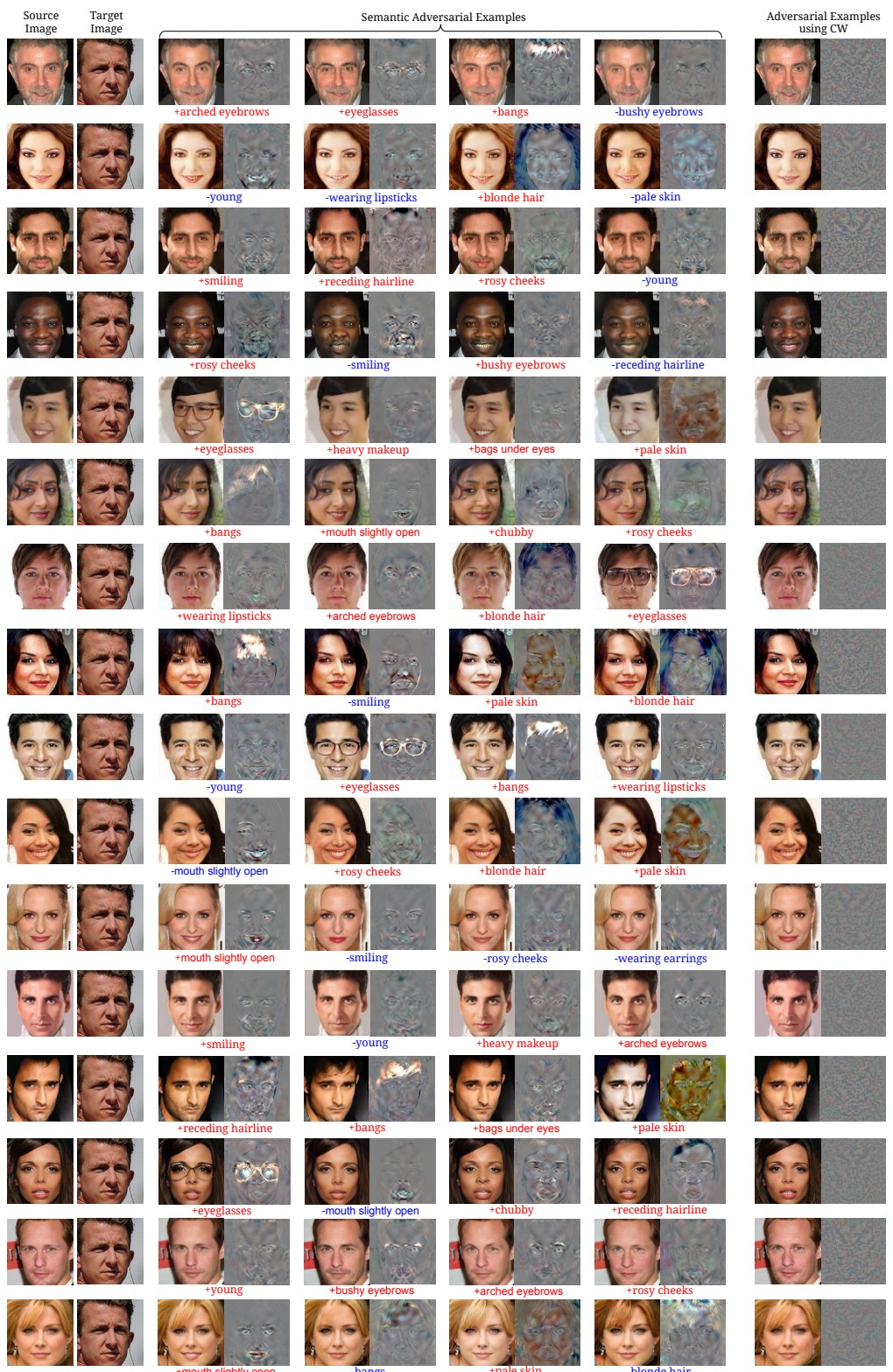

Figure N: Qualitative comparisons between our proposed *SemanticAdv* (G-FPR = $10^{-3}$) and pixel-wise adversarial examples generated by CW. Along with the adversarial examples, we also provide the corresponding perturbations (residual) on the right.

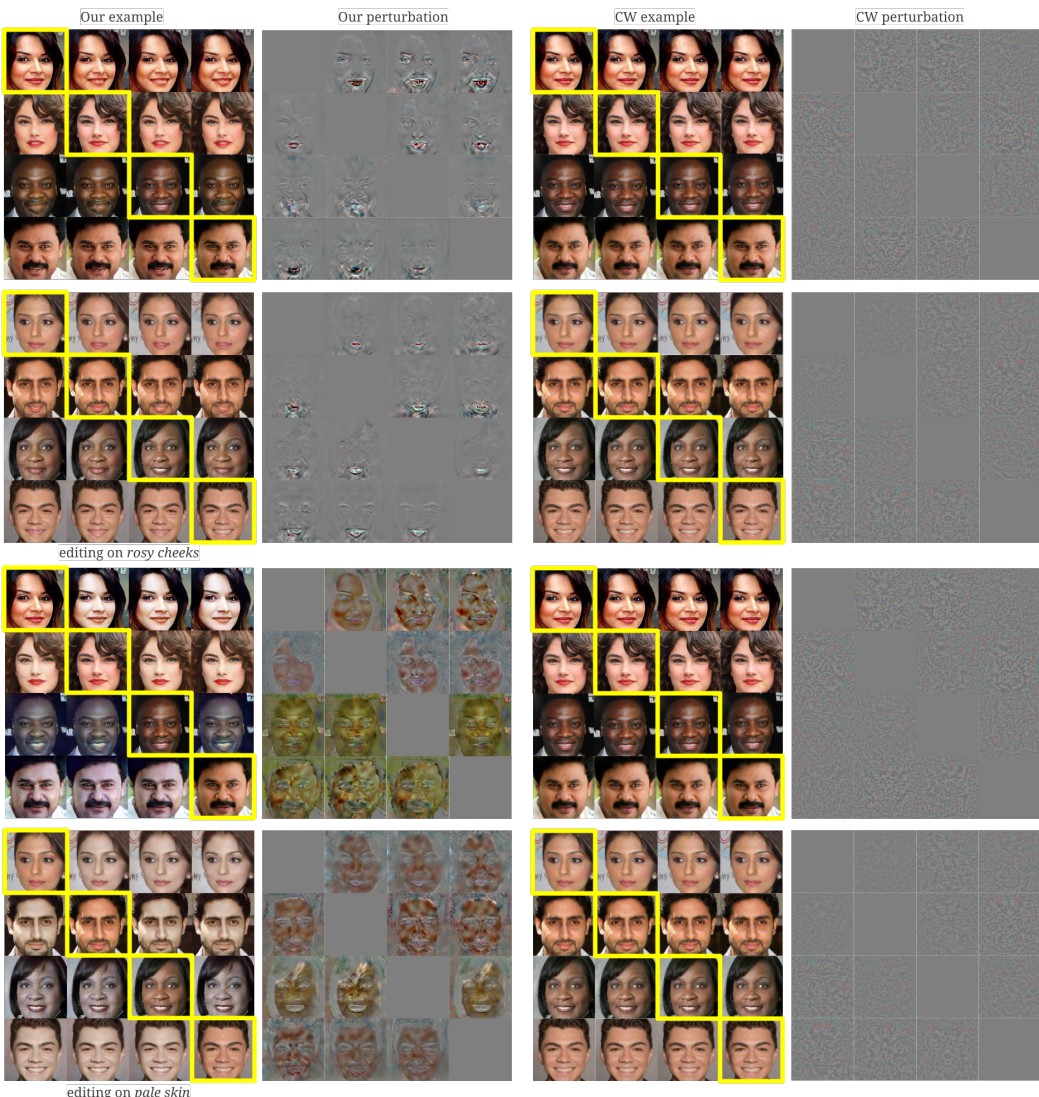

Figure O: Qualitative analysis on single-attribute adversarial attack (*SemanticAdv* with G-FPR = $10^{-3}$) by each other. Along with the adversarial examples, we also provide the corresponding perturbations (residual) on the right.

