# OpenReview forum: "SemanticAdv: Generating Adversarial Examples via Attribute-Conditional Image Editing"
_ICLR.cc/2020/Conference — Reject_

### Official Review · AnonReviewer2 · 2019-10-19
**Official Blind Review #2**

**Rating:** 6

**Review:**

Summary:
This paper proposes to generate "unrestricted adversarial examples" via attribute-conditional image editing. Their method, SemanticAdv, leverages disentangled semantic factors and interpolates feature-map with higher freedom than attribute-space. Their adversarial optimization objectives combine both attack effectiveness and interpolation smoothness. They conduct extensive experiments for several tasks compared with CW-attack, showing broad applicability of the proposed method.

The paper is well written and technically sound with concrete experimental results. I'm glad to suggest accepting the paper.

With the help of attribute-conditional StarGAN, SemanticAdv generates adversarial examples by interpolating feature-maps conditioned on attributes. They design adversarial optimization objectives with specific attack objectives for identity verification and structured prediction tasks. They provide experiments showing the effectiveness of SemanticAdv; analysis on attributes, attack transferability, black-box attack, and robustness against defenses; as well as user study with subjective. The qualitative results also look nice and the code base is open-sourced.

A question out of curiosity, the last conv layer in the generator is used as the feature-map. How is the attack effectiveness of using other layers?

**Experience Assessment:**

I have published one or two papers in this area.

**Review Assessment: Checking Correctness Of Derivations And Theory:**

I assessed the sensibility of the derivations and theory.

**Review Assessment: Checking Correctness Of Experiments:**

I assessed the sensibility of the experiments.

**Review Assessment: Thoroughness In Paper Reading:**

I read the paper at least twice and used my best judgement in assessing the paper.

---

> ### Author Response · Authors · 2019-11-14
> **Response to Review #2**
>
>
> We thank the reviewer for the constructive suggestions and comments, and we have conducted additional experiments based on the comments.
>
> Q1. Effectiveness of using other layers?
>
> We have tested another two feature maps ($f_{1}$, $f_{2}$) after the first/second up-sampling operations as shown in Table E (see Section D in our appendix) in the submitted paper; and we also conducted additional experiments on two extra feature maps ($f_{-2}$, $f_{-1}$) based on the suggestions. $f_{-2}$ indicates the first feature map after the last down-sampling operations and $f_{-1}$ represents the feature map after $f_{-2}$. The full results are shown in the revision Table E and F.
> We also present the results as below. The result shows that samples generated by interpolating on our selected layer ($f_0$) achieve the highest attack success rate.
>
> +──────────+───+───+───+───+───+───+───+───+───+
> | T-FPR(G-FPR)              |          $10^{−3}(10^{−3})$       |  $3\times10^{−3}(3\times10^{−3})$ |         $10^{−4}(10^{−4})$        |
> +──────────+───+───+───+───+───+───+───+───+───+
> | Layer(f)                        | $f_{-2}$    | $f_{-1}$    | $f_0$      | $f_{-2}$    | $f_{-1}$    |   $f_{0}$    | $f_{-2}$    | $f_{-1}$    | $f_{0}$      |
> +──────────+───+───+───+───+───+───+───+───+───+
> | Attack Success Rate  | 49.4   | 92.09 | 99.29  | 30.44 | 81.87 | 97.35 | 6.66   | 45.46 | 76.64 |
> +──────────+───+───+───+───+───+───+───+───+───+

---

### Official Review · AnonReviewer3 · 2019-10-19
**Official Blind Review #3**

**Rating:** 3

**Review:**

The authors describe a method for adversarially modifying a given (test) example that 1) still retains the correct label on the example, but 2) causes a model to make an incorrect prediction on it. The novelty of their proposed method is that their adversarial modifications are along a provided semantic axis (e.g., changing the color of someone's skin in a face recognition task) instead of the standard $L_p$ perturbations that the existing literature has focused on (e.g., making a very small change to each individual pixel). The adversarial examples that the authors construct, experimentally, are impressive and striking. I'd especially like to acknowledge the work that the authors put in to construct an anonymous link where they showcase results from their experiments. Thank you!

Overall, I think that this is interesting work that can help to broaden the study of adversarial examples and make them more applicable even in non-adversarial settings (e.g., by making models more robust to the changes in semantic attributes that the authors consider). There has been quite a bit of interest in the community in adversarial examples that are not just $L_p$ perturbations, and I believe that the authors' approach will encourage a good deal of follow-up research.

However, my main concern with the paper is that in my opinion, it does not sufficiently address why it is important to generate adversarial examples in the way they do. For example:

1) Is the argument that this is a more powerful attack surface, so adversaries should take note (and defenders should figure out how to defend against this)? If that is the case, what is the attack model under which these attacks are realistic? For example, the original $L_\infty$ attacks are motivated in the sense that the adversarial examples are visually imperceptible, so they might not be noticed by the end-user. What is the equivalent argument for these semantic attacks?

2) Is the argument that these semantic attacks somehow capture a more realistic part of the data distribution over all natural images, and therefore it is good to have models that perform well on these semantic adversarial examples even if we're not concerned about an adversary (e.g., because the model might generalize better to other tasks or be more causally correct)? If that's the case, then I think this needs to be explored more. For example, what about the following straw man baseline: use a controllable semantic-attribute-based generator to generate semantically different images without any notion of an adversarial attack, and then do standard $L_p$ attacks on that generated image? How would that be better or worse than the proposed method?

3) Or is the argument that it is just good to be able to generate examples that models get wrong? If so, why, and why is this method better than other methods?

I think the paper would be significantly stronger if the importance and implications of their work were explicated along the above lines. For this reason, my current assessment is a weak reject, though I'd be open to changing this assessment.

=== Less critical comments, no need to respond or fix right away ===

While the overall concept and approach was clear, I generally found the notation and mathematical exposition difficult to follow. Please be more precise. Here is a non-exhaustive list of examples from section 3:

a) I'm not sure what's the difference between $x^\text{tgt}$ and $x^\text{adv}$, or between $x^\text{new}$ and  $x^*$. These seem to be used somewhat interchangeably?

b) Equation 3 is the central optimization problem in the paper, and should be written out explicitly using $\alpha$ as the optimization variable, instead of referring to equations 1 and 2 (in which $x^*$ doesn't even appear).

c) I didn't understand equation 4. What does assuming $M(x^\text{tgt}) = y^\text{tgt}$ mean? What happens when that is not true?

d) Equation 5: Why is $y$ in the right hand side by not in the left?

e) Equation 6: $L_\text{smooth}$ is missing an argument.





**Experience Assessment:**

I have read many papers in this area.

**Review Assessment: Checking Correctness Of Derivations And Theory:**

N/A

**Review Assessment: Checking Correctness Of Experiments:**

I assessed the sensibility of the experiments.

**Review Assessment: Thoroughness In Paper Reading:**

I read the paper at least twice and used my best judgement in assessing the paper.

---

> ### Author Response · Authors · 2019-11-14
> **Response to Review #3**
>
>
> We appreciate the reviewer’s precious comments and suggestions. We thank the reviewer for recognizing our work as helpful to broaden the study of adversarial examples and encourage a good deal of follow-up research. We will first provide the high-level motivation of why we need to generate adversarial examples and then answer the individual questions. We have revised the notations and equations in our updated manuscript.
>
> Q1. Why it is important to generate adversarial examples in the way they do?
>
> A1: Thanks for the question, the reasons/motivations are described below.
>
> Deep Neural Networks (DNNs) have achieved great success in a variety of applications. However, various security threats are emerging with the deployment of machine learning models. Without a deep understanding of how neural networks fail under attacks, it would be concerning to apply them in security-critical systems such as face verification and autonomous driving systems. Additionally, learning systems are usually required to be immune to *reasonable variations* of the input.
>
> So far, such *variations* have been focused on imperceptible perturbation added to the given inputs whose magnitude is bounded by pixel-space $L_p$-norm. Some works have discussed the limitations of only measuring and evaluating the $L_p$ bounded perturbation [a1,a3,a4]. Therefore, it is important to explore other non-$L_p$ bounded perturbation, especially semantically meaningful perturbation, and more detailed reasons are listed below.
>
> First, the semantic based perturbation is new and interesting, which contains different intrinsic properties compared with the traditional $L_p$ bounded attacks. For instance, the semantic perturbation could be very large to cover the other side of $L_p$ bounded perturbation.
>
> Second, in our proposed semantic based adversarial examples, we can explicitly control the desired editing attribute (e.g. hair color), and successfully preserve the high perceptual quality of the generated images as shown in Figure 4. This would help to explore the vulnerability/sensitivity of different semantic attributes.
>
> Third, various methods have been proposed to defend against adversarial attacks. Adversarial training based methods are currently the most efficient. Currently most adversarial training methods are only effective against a small set of seen attacks [a1], and researchers (e.g., Kang, et. al. [a2]) have shown that generating diverse attacks can help improve adversarial training performance against unseen attacks. Therefore, we believe that our semantic adversarial examples can potentially benefit adversarial training to improve model robustness by providing diverse unseen adversarial examples.
>
> In addition, partially based on the reasons above, Brown, et. al.[a4] proposed the unrestricted adversarial example challenge to encourage the community to explore the adversarial space beyond $L_p$, which would potentially benefit the adversarial learning research, and we do hope SemanticAdv can contribute as well.
>
>
> (To be continued.)

---

> > ### Author Response · Authors · 2019-11-14
> > **Continue #1**
> >
> >
> > Q2. Is the argument that this is a more powerful attack surface, so adversaries should take note (and defenders should figure out how to defend against this)?
> > A2: Thanks for the interesting question. There are several definitions for “more powerful attack”. For instance, in terms of the magnitude of perturbation, SemanticAdv is more powerful as it is able to tolerate a larger perturbation. In terms of attack success rate, both SemanticAdv and other $L_p$ norm based pixel level attacks can achieve almost 100% attack success rate. Therefore, we believe semantic based adversarial examples are important mainly because it explores different properties that traditional $L_p$ based ones have missed and provide diverse adversarial attacks.
> >
> > Q3. Capture a more realistic part of the data distribution over all natural images.
> > A3: Thanks for the very interesting point. At this point, we can only show that such semantic based attacks do provide diverse adversarial examples in addition to existing ones, but whether it actually captures more realistic part of the data distribution is challenging to verify and we will definitely explore it as the future work by proposing different evaluation process and metrics for the benign and adversarial data distributions.
> >
> > Based on the reviewer’s suggestion, we conduct the StawnMan baseline. We generate adversarial examples by using StawnMan baseline. It shows 100% attack success rate under the white-box setting. We further evaluate its performance of query-free black-box API attacks and transferability. The results are shown below. We can observe there is a noticeable gap between our proposed SemanticAdv and the StawnMan baseline in terms of performance. This result justifies the argument that our SemanticAdv is able to produce novel adversarial examples that cannot be simply achieved by combining attribute-conditional image editing model with $L_p$ bounded perturbation.
> >
> > Table R3T1: Transferability of StawnMan
> > +────────+───────+
> > | M_test / M_opt   | R-101-S             |
> > +────────+───────+
> > | R-50-S                  | 0.035 (0.108)    |
> > | R-101-S                | 1.000 (1.000)    |
> > | R-50-C                  | 0.145 (0.202)    |
> > | R-101-C                | 0.085 (0.236)    |
> > +────────+───────+
> > (G-FPR = $10^{-3}$, T-FPR = $10^{-3}$, SemanticAdv in blankets)
> >
> > +────────+───────+
> > | M_test / M_opt   | R-101-S             |
> > +────────+───────+
> > | R-50-S                  | 0.615 (0.862)    |
> > | R-101-S                | 1.000 (1.000)    |
> > | R-50-C                  | 0.570 (0.837)    |
> > | R-101-C                | 0.695 (0.888)    |
> > +────────+───────+
> > (G-FPR = $10^{-4}$, T-FPR = $10^{-3}$, SemanticAdv in blankets)
> >
> > Table R3T2: Quantitative analysis on query-free black-box attack of StawMan
> >
> > +──────────────+──────────────+──────────────+
> > | API name                                   |                   Face++                       |                     AliYun                     |
> > +──────────────+──────────────+──────────────+
> > | Attacker / Evaluation Metric   | T-FPR = $10^{-3}$  | T-FPR = $10^{-4}$ | T-FPR = $10^{-3}$  | T-FPR = $10^{-4}$ |
> > +──────────────+──────────────+──────────────+
> > | StawnMan (G-FPR = 1e-3)        | 10.71                | 4.08                 | 3.00                  | 0.00                 |
> > | SemanticAdv (G-FPR = 1e−3)   | 27.32                | 9.79                 | 7.50                  | 2.00                 |
> > | StawnMan (G-FPR = 3e-4)        | 21.32                | 9.14                 | 7.50                  | 1.50                 |
> > | SemanticAdv (G-FPR = 3e−4)   | 57.22                | 38.66               | 29.50                | 17.50               |
> > | StawnMan (G-FPR < 1e-4)        | 27.69                | 15.38               | 10.00                | 3.00                 |
> > | SemanticAdv (G-FPR < 1e−4)   | 64.63                | 42.69               | 35.50                | 22.17               |
> > +──────────────+──────────────+──────────────+
> >
> >
> >
> >
> > Q4. It is just good to be able to generate examples that models get wrong? If so, why, and why is this method better than other methods?
> >
> > A4: According to the evaluation in our paper and the addition StawnMan baseline results, it shows that SemanticAdv is not only effective to generate adversarial examples different with $L_p$ based attacks but also indeed contain unique properties (e.g. different from applying $L_p$ perturbation on the manipulated image guided by semantic attributes). One of the potential reasons for the good performance is that SemanticAdv is able to explore the stronger adversarial space which can achieve higher transferability. For more detailed motivation of the semantidAdv please refer to A1.
> >
> >
> > (To be continued.)

---

> > > ### Author Response · Authors · 2019-11-14
> > > **Continue #2**
> > >
> > >
> > > Q5. Notations.
> > > Q5a: The difference between $x^{tgt}$ and $x^{adv}$, or between $x^{new}$ and $x^{*}$.
> > > A5a: We consider the adversarial attack in the targeted setting, where $x$ is the original image and $x^{tgt}$ is the image with the target label we aim at misclassifying $x^{adv}$ to.
> > > $x^{new}$ is the intermediate image we produced by the attribute-conditional image editing from the original $x$ (without adversarial attack).
> > > $x^{*}$ represents the intermediate image in the optimization step (e.g., $x^{*}$ equals to $x^{adv}$ at the end of optimization).
> > >
> > > Q5b: Equation 3.
> > > A5b: Thanks for the suggestion. We will improve the equation 3 denoting with optimization variable alpha.
> > >
> > > Q5c: $M(x^{tgt}) = y^{tgt}$
> > > A5c: Thanks for pointing this out and we realize it causes confusion and will remove this notation in the revision. Basically, this notation is used to guarantee the unperturbed instances evaluated by our algorithms can be predicted correctly by $M$ otherwise it would be a challenge to distinguish the source of the error of the generated instances.
> > >
> > > Q5d: Position of y.
> > > A5d: Thanks for pointing it out! It is a typo, it should be $y^{*}$. We will fix it in the revision.
> > >
> > > Q5e: Missing argument of $L_{smooth}$.
> > > A5e: We admit that this is an abbreviated form of $L_{smooth}(\alpha)$, which has been defined in Equation (3). We will update this in our revision.
> > >
> > > References
> > > [a1] “Spatially transformed adversarial examples.” Xiao et al. In ICLR 2018.
> > > [a2] “Testing robustness against unforeseen adversaries.” Kang et al. arXiv preprint arXiv:1908.08016.
> > > [a3] “Wasserstein Adversarial Examples via Projected Sinkhorn Iterations.” Wong et al. In ICML 2019
> > > [a4] “Unrestricted adversarial examples.” Brown et al. arXiv preprint arXiv:1809.08352.

---

### Official Review · AnonReviewer1 · 2019-10-21
**Official Blind Review #1**

**Rating:** 6

**Review:**

This paper proposes adversarial attacks by modifying semantic properties of the image. Rather than modifying low-level pixels, it modifies mid-level attributes. The authors show that the proposed method is effective and achieves stronger results than the pixel-level attack method (CW) in terms of attacking capability transferring to other architectures. Importantly, the authors show results on a variety of tasks, e.g. landmark detection and segmentation in addition to classification/identification. The most related work is Joshi 2019 and the authors show that the method used in that work (modification in attribute space) is inferior to modification in feature space still via attributes, as the authors proposed. However, I have a few comments and concerns:
1) The authors mention on page 3 they assume M is an oracle-- what is the impact of this?
2) The results in Table C don't look good-- the proposed method can *at best* (in a generous setup) equal the results of CW-- maybe I missed something but more discussion would be helpful.
3) Is there a way to evaluate the merits of semantic modification (beyond attack success) in addition to "does it look reasonable"? The authors mention attribute-based modifications are more practical, how can this be evaluated? If attribute-based attacks are better, is there a cost to this? How easy is it to make attribute-based attacks compared to low-level ones?
4) The authors mention that for their transferrability results, they "select the successfully attacked..." (page 7). What is the impact of this, as opposed to selecting non-successfully attacked samples?
5) Re: behavior with defense methods, is the advantage of the proposed method a matter of training the defense methods in a tailored way, so they're aware of attribute-based attacks?

**Experience Assessment:**

I have read many papers in this area.

**Review Assessment: Checking Correctness Of Derivations And Theory:**

I assessed the sensibility of the derivations and theory.

**Review Assessment: Checking Correctness Of Experiments:**

I carefully checked the experiments.

**Review Assessment: Thoroughness In Paper Reading:**

I read the paper thoroughly.

---

> ### Author Response · Authors · 2019-11-14
> **Response to Review #1**
>
>
> We really appreciate the reviewer’s precious comments. Sorry for the potential confusion. We would like to answer your questions as follows and we have added them in our revision.
>
> Q1: assume M is an oracle-- what is the impact of this?
> A1: Thanks for pointing this out and we will remove this notation in the revision to avoid confusion. Basically, M here is used to obtain the corresponding label related to data x, and we actually do not need to use this assumption in our experiments (we can assume the ground-truth label is given). But we see this assumption introducing the confusion and we will remove this statement by using the ground truth label y directly.
>
> Q2: “The results in Table C don't look good.”
> A2: We believe the “Table C don’t look good” refers to the results with “worst” and “average” metrics. In Table C, the “best” metric of SemanticAdv should be served as a fair comparison to CW, where both methods achieve 100% attack success rate. Therefore, our result is good. The detailed reasons are as follows.
>
> For each victim image, our SemanticAdv generates a total of 17 adversarial images by augmenting one semantic attribute each time (e.g., we have 17 attributes to manipulate). However, CW generates a single adversarial example regardless of attributes, which can be viewed as instance-level generation. Therefore, we compare CW with our SemanticAdv on the instance-level which corresponds to the “best” metric.
>
> In addition, we report the performance using the “average” and “worst” metric, which actually provides additional insights into the robustness of face verification models across different attributes. Combining the results from Table C in our appendix and Figure 3, we understand that the face verification models used in our experiments have different levels of robustness across attributes. For example, face verification models are more robust against local shape variations than color variations, e.g., pale skin has higher attack success rate than mouth open. We believe these discoveries will help the community further understand the properties of face verification models.
>
> To summarize, our *semantic* adversarial examples not only achieves attack success rate comparable to traditional $L_p$-norm bounded CW attacks, but also enables us to investigate the model robustness under different semantic attributes. We will make the description of Table C clearer in the revised manuscript.
>
>
> (To be continued.)

---

> > ### Author Response · Authors · 2019-11-14
> > **Continue**
> >
> >
> > Q3a: Is there a way to evaluate the merits of semantic modification (beyond attack success) in addition to “does it look reasonable”?
> > A3a: As far as we know, user study has been widely used in the literature when it comes to the qualitative measurement of adversarial examples [a1]. Other measurement such as $L_p$ bound has been known drawbacks and limitations as discussed in [a1, a2, a3]. We admit it is non-trivial to devise perceptual metrics to measure the perceptual quality of adversarial samples in a systematic manner, and it is truly a challenging open problem in the vision and learning community.
> >
> > Q3b:The authors mention attribute-based modifications are more practical, how can this be evaluated?
> > A3b: Thanks for the question and sorry for the confusion. In our scenario, “more practical” means it is relatively easier for someone to realize semantic attributes in practice than perform $L_p$ based perturbation. For instance, one can realize the semantic attribute editing to the faceID system by wearing a pair of glasses or have the hair dyed with a different color.
> >
> > Q3c: If attribute-based attacks are better, is there a cost to this?
> > A3c: Thanks for the interesting question! The extra costs of SemanticAdv are from two sources: (1) we need the corresponding attribute annotation for each image or a pre-trained attribute classifier to predict the attribute labels; and (2) we need to train a generative model to conduct attribute-conditional image editing.
> > These two problems happen to be popular research topics in the vision and learning community with tremendous progress in the past few years, which we can leverage.
> >
> > Q3d: How easy is it to make attribute-based attacks compared to low-level ones?
> > A3d: First, we observe that generating the attribute-based attacks is as efficient as the low-level ones. We conduct additional experiments to evaluate the running time. The detailed setting can be found in Section A. It takes  on average 0.30s for CW to generate a single adversarial example on single GTX 1080Ti while the running time is 0.32s for our SemanticAdv. Besides the efficiency, we believe the model optimization of SemanticAdv is as easy as CW attack, except that SemanticAdv requires a pre-trained attribute-conditional image generation model available (see Q3c).
> >
> > Q4: Impact of “selecting successfully attacked example” to evaluate the transferability.
> > A4: This is the standard setting in the literature [a1] when it comes to attack transferability evaluation. We will make this clear in the revision.
> >
> > Q5: The advantage of the proposed method.
> > A5: Thanks for pointing this out. Our proposed method has three major advantages:
> > (1) SemanticAdv helps identify specific semantic-based adversarial examples for a machine learning model (e.g., face verification network, scene segmentation network) to further explore corner cases in the representation; (2) as the reviewer points out, such semantic-based attacks can enlarge the diversity of seen adversarial examples and therefore help improve model robustness by training with them against unseen ones as discussed in [a4]; and (3) analyzing the defense effectiveness with SemanticAdv by modifying different attributes could help better understand the model vulnerabilities from semantic perspective.
> >
> > References
> > [a1] “Spatially transformed adversarial examples.” Xiao et al, In ICLR 2018.
> > [a2] “Wasserstein Adversarial Examples via Projected Sinkhorn Iterations.” Wong et al. In ICML 2019
> > [a3] “Unrestricted adversarial examples.” Brown et al. arXiv preprint arXiv:1809.08352 2018.
> > [a4] “Testing robustness against unforeseen adversaries.” Kang et al. arXiv preprint arXiv:1908.08016 2019.

---

### Public Comment · ~Anthony_Wittmer1 · 2019-10-08
**A closely related paper**

Great work and I really enjoy reading it.

However, previous work has also studied the semantic attack to fool models. Please check out this paper [1]. For the attack of semantic attributes, to my knowledge, [1] is the first work to perform the semantic attack to fool DNNs by designing specific eyeglasses.

In my opinion, a discussion/comparison seems due.

[1] Accessorize to a Crime: Real and Stealthy Attacks on State-of-the-Art Face Recognition

---

> ### Author Response · Authors · 2019-10-09
> **reply to "A closely related paper"**
>
> Thanks for the nice reference!
> We mainly valued the mentioned paper based on its physical attack effectiveness, but we will definitely add this interesting discussion about it!

---

### Public Comment · ~Zhenhua_Chen1 · 2019-10-23
**Interesting work**

Hi,

The idea of semantic manipulation is pretty interesting. I wonder how these perturbations generalize? For example in Figure 1 Right, the perturbation is supposed to make a face pale and fool a face verification system. Would the same perturbation still work for a different face?

---

> ### Author Response · Authors · 2019-10-27
> **Reply to "generalization of semantic perturbations".**
>
> Thank you for your interest!
>
> In our experiments, we observe that our proposed semantic perturbation is generalizable to some extent. Specifically, the same type of semantic perturbation can be applied to attack different samples (faces). Also, feel free to check our anonymous website where we show good examples of applying the same type of perturbations (e.g., young --> senior, regular skin --> pale skin) to synthesize adversarial face images against the verification model (see Figure 2 on the anonymous website).
>
> https://sites.google.com/view/generate-semantic-adv-example

---

### Author Response · Authors · 2019-11-15
**Responses and Revisions**


We thank all reviewers for their valuable comments and suggestions.  We appreciate the reviewers recognizing our work interesting (R1, R2, R3), technically sound with concrete experiment results (R2, R3), broadening the study of adversarial examples and encouraging a good deal of follow-up research (R3). Based on the reviewers’ suggestions, we have made the following changes in our revision.
    1. Adding StawnMan baseline proposed by R3 in Table H and I.
    2. Selecting additional different layer’s feature map for interpolation and evaluating the results. (Table F)
    3. Changing the notations of the equations in Section 3.
    4. Fixing some typos.

---

### Decision · Program_Chairs · 2019-12-19

**Decision:**

Reject

**Comment:**

I had a little bit of difficulty with my recommendation here, but in the end I don't feel confident in recommending this paper for acceptance, with my concerns largely boiling down to the lack of clear description of the overall motivation.

Standard adversarial attacks are meant to be *imperceptible* changes that do not change the underlying semantics of the input to the human eye. In other words, the goal of the current work, generating "semantically meaningful" perturbations goes against the standard definition of adversarial attacks. This left me with two questions:

1. Under the definition of semantic adversarial attacks, what is to prevent someone from swapping out the current image with an entirely different image? From what I saw in the evaluation measures utilized in the paper, such a method would be judged as having performed a successful attack, and given no constraints there is nothing stopping this.

2. In what situation would such an attack method would be practically useful?

Even the reviewers who reviewed the paper favorably were not able to provide answers to these questions, and I was not able to resolve this from my reading of the paper as well. I do understand that there is a challenge on this by Google. In my opinion, even this contest is somewhat ill-defined, but it also features extensive human evaluation to evaluate the validity of the perturbations, which is not featured in the experimental evaluation here.

While I think this work is potentially interesting, it seems that there are too many open questions that are not resolved yet to recommend acceptance at this time, but I would encourage the authors to tighten up the argumentation/evaluation in this regard and revise the paper to be better accordingly!